# Speed-Dependent Bearing Models for Dynamic Simulations of Vertical Rotors

**Gudeta Berhanu Benti [1,\*], Rolf Gustavsson [2] and Jan-Olov Aidanpää [1]**

1 Department of Engineering Sciences and Mathematics, Luleå University of Technology, 971 87 Luleå, Sweden; jan-olov.aidanpaa@ltu.se
2 Vattenfall AB Research and Development, 814 26 Älvkarleby, Sweden; rolf.gustavsson@vattenfall.com
\* Correspondence: gudeta.benti@ltu.se

**Abstract:** Many dynamic simulations of a rotor with a journal bearing employ non-linear fluid-film lubrication models and calculate the bearing coefficients at each time step. However, calculating such a simulation is tedious and computationally expensive. This paper presents a simplified dynamic simulation model of a vertical rotor with tilting pad journal bearings under constant and variable (transient) rotor spin speed. The dynamics of a four-shoes tilting pad journal bearing are predefined using polynomial equations prior to the unbalance response simulations of the rotor-bearing system. The Navier–Stokes lubrication model is solved numerically, with the bearing coefficients calculated for six different rotor speeds and nine different eccentricity amplitudes. Using a MATLAB inbuilt function (*poly53*), the stiffness and damping coefficients are fitted by a two-dimensional polynomial regression and the model is qualitatively evaluated for goodness-of-fit. The percentage relative error (*RMSE%*) is less than 10%, and the adjusted R-square ($R^2_{adj}$) is greater than 0.99. Prior to the unbalance response simulations, the bearing parameters are defined as a function of rotor speed and journal location. The simulation models are validated with an experiment based on the displacements of the rotor and the forces acting on the bearings. Similar patterns have been observed for both simulated and measured orbits and forces. The resultant response amplitudes increase with the rotor speed and unbalanced magnitude. Both simulation and experimental results follow a similar trend, and the amplitudes agree with slight deviations. The frequency content of the responses from the simulations is similar to those from the experiments. Amplitude peaks, which are associated with the unbalance force ($1 \times \Omega$) and the number of pads ($3 \times \Omega$ and $5 \times \Omega$), appeared in the responses from both simulations and experiments. Furthermore, the suggested simulation model is found to be at least three times faster than a classical simulation procedure that used FEM to solve the Reynolds equation at each time step.

**Keywords:** vertical rotor; rotor dynamics; transient; tilting pad journal bearing; bearing coefficient

## 1. Introduction

Tilting pad journal bearings (TPJBs) are used in many different applications, being preferred for their stability and ease of service. Many studies have been carried out on the dynamics of TPJBs, with Lund [1], in 1964, the first researcher to present the dynamic characteristics of 4 shoes, 5 shoes, 6 shoes, and 12 shoes TPJBs by introducing the so-called "pad assembly method". Over the past 50 years, TPJBs have been investigated both theoretically and experimentally, and research by Nicholas et al. [2], Jones et al. [3], and Keith [4] was among the earlier work to study the effect of various TPJB parameters on the dynamic bearing coefficients. Other researchers, such as Lund et al. [5,6] and Someya [7], made a remarkable contribution to journal bearing design and presented the eight stiffness and damping coefficients of different types of journal bearings (including TPJBs), which are still being used today as a design guideline for many journal bearings. Besides, their scientific work and findings contributed to the advancement of modern theoretical models.

Recent theoretical studies dealing with the dynamics of both fixed geometry and TPJBs use advanced modeling by taking into account many different factors. These include mechanical deformation [8–10], thermal effect [8,11,12], pad flexibility [13–15], and pivots flexibility [16–18]. Dimond et al. [19] carried out an extensive review of the development of TPJB analysis. The paper reviewed the earlier research into thermohydrodynamic (THD), thermoelastohydrodynamic (TEHD), and bulk-flow analyses. Besides, the development of synchronous and non-synchronous bearing models was discussed. The excitation frequency effect on the stiffness and damping coefficients of TPJBs has been a long-standing discussion topic for many years. Several researchers [20–22] have investigated the issue and described their findings based on theoretical and/or experimental results.

In rotordynamics, a reliable fluid film-bearing model estimation is important for the accuracy of the predicted dynamics of a system. In the advanced bearing models, the fluid film lubrication model, which is represented by Reynolds or Navier–Stokes equations, is solved numerically. Many numerical computations employ nonlinear lubrication models to calculate the dynamic coefficients of the journal bearing in both horizontal and vertical installations [23–26]. This means the fluid film-bearing model has to be solved at each time step simultaneously with the rotor-bearing system simulations. White et al. [27] and Cha et al. [25] calculated the nonlinear bearing forces by solving the Reynolds equation at each time step. This is, however, a time-consuming and computationally expensive process. Many attempts have been made by researchers to simplify and improve the computation efficiency of the simulation procedure. Perez et al. [28] used a simplified analytical bearing force expression to model the dynamics of a vertical rotor with hydrodynamic journal bearing. This approach has previously been presented by Yu Huang et al. [29] to estimate the instability threshold speed of cylindrical journal bearings. Nässelqvist et al. [30] suggested a method to make the computation faster by predefining the bearing coefficients with polynomials as a function of journal eccentricity. Childs et al. [31] and Tschoepe et al. [32] investigated and compared the performance of the TPJBs in the load-on-pad (LOP) and load-between-pad (LBP) configurations. Nässelqvist et al. [30] approximated bearing coefficients using sinusoidal equations over the range of journal radial positions. Synnegård et al. [33] and Benti et al. [34] described how the TPJBs can excite the system and cause vibration at certain frequencies due to the number of pads.

The present work proposes a simplified and computationally efficient model for calculating the dynamic response simulation of a rotor with journal bearings. It is an extension of the previous study [30] that used one-dimensional polynomial equations to represent the bearing coefficients as a function of journal eccentricity. One drawback of the previous model is that it does not take into account the speed-dependency of the bearing characteristics, and the polynomial equations apply only to a particular rotor speed. If the rotor speed changes, the polynomial equations have to be recalculated. For this reason, the method is limited and impractical for response simulations under variable speed. In the proposed model, however, the bearing coefficients are represented by two-dimensional polynomial equations as a function of journal center location and rotor speed. The bearing model is general and can be used for a wide range of applications. One limitation of the proposed model is the preliminary computational effort required to develop the bearing models.

In this paper, the dynamic response of a rigid rotor supported by two four-shoe TPJBs was studied. No static radial load was considered, and the bearings were subjected to a dynamic load due to a rotating unbalance. The proposed dynamic model was evaluated in terms of accuracy and computational time efficiency. Validation of the model was made by comparing the trajectory of the journal center and bearing reaction forces with the corresponding experimental results. Furthermore, the computational efficiency of the model (in terms of simulation time) was compared with a classical simulation model that simultaneously computed the fluid film lubrication model at each time step.

## 2. Experiment

### 2.1. Rotor Rig Description

An experimental test was carried out with a rotor rig installed vertically, as shown in Figure 1. A symmetric steel rotor was supported by two identical four-shoe TPJBs at the top and bottom end of the rotor. Each bearing was attached to the structure of the rotor rig via a bracket. The stiffness of the bracket was estimated to be 500 MN/m [35]. As shown in Figure 2, the bearings and brackets were connected in series. The rotor was suspended vertically and driven by an electric motor. A slender stinger was used to link the electric motor to the rotor and provided support only in the axial direction. It was attached to two jaw couplings at both ends to minimize the transfer of torque due to misalignments. An unbalanced mass was attached to the rotor at a distance of 70 mm from the axis of the rotor. The rotor and the brackets were intentionally designed to be stiffer than the bearings in order to be able to investigate the dynamics of the bearings.

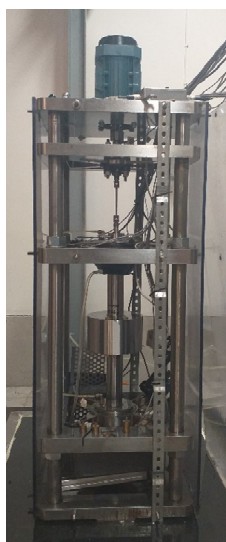

**Figure 1.** Rotor rig.

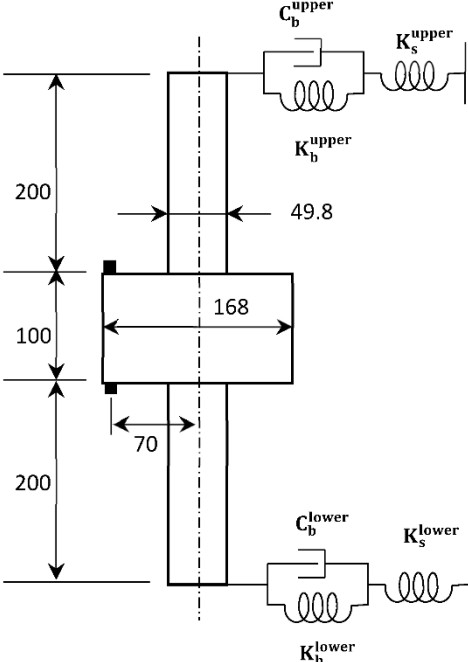

**Figure 2.** Schematic representation of the rotor rig. All dimensions are given in mm.

The displacement of the rotor at each bearing was measured by four 4 mm inductive proximity displacement sensors (Contrinex DW-AD-509-M8) mounted on the bearing housing. The sensors were calibrated on-site for specific target material, which is steel in this case. Furthermore, a full Wheatstone bridge strain gauge (Kyowa KFG-5-350-D16-11L3M2S) was mounted on each bracket to measure the bearing reaction forces. Similarly, the sensors were calibrated on-site both directly (shunt calibration) and indirectly (by applying a known force). Angular speed of the rotor was measured by an optical sensor with about ±1 rpm accuracy. All sensors were calibrated on-site by certified calibration equipment. A universal MX840 amplifier of HBM Quantum data acquisition system and Catman data acquisition software were used for measurement. The technical specification of the rotor rig is given in Table 1, and a further description is available in [35].

**Table 1.** Technical specification of the rotor rig.

| Descriptions | Values |
|---|---|
| Rotor diameter (mm) | 49.84 |
| Rotor length (mm) | 500 |
| Disk diameter (mm) | 100 |
| Disk thickness (mm) | 168 |
| Direction of rotation | Counterclockwise |
| The stiffness of the bracket (MN/m) | 500 |
| Rotor mass (kg) | 24.74 |

### 2.2. Bearing Description

The TPJBs had four identical pads (with rocker pivots), located at $0°$, $90°$, $180°$, and $270°$ from the *x*-axis. Each pad was made of a 1 mm thick layer of white metal (babbitt) lining and backed with a 7 mm thick layer of steel. Figure 3 and Table 2 show the schematic representation and a detailed specification of the four-shoe TPJB, respectively. The parameters were chosen so that the Sommerfeld number of the bearings resembles those in hydropower generators. The bearings were supplied with 0.01 MPa lubricant (Q6 Handel oil) and operated under fully flooded lubrication conditions. During each test, the inlet and outlet lubrication temperatures were measured by PT100 thermocouples (with ±0.5 °C accuracy). The temperature sensors were installed right before and after the upper and lower TPJBs. As shown in Figure 4, the inlet and outlet average temperature measurements are plotted as a function of rotor speed.

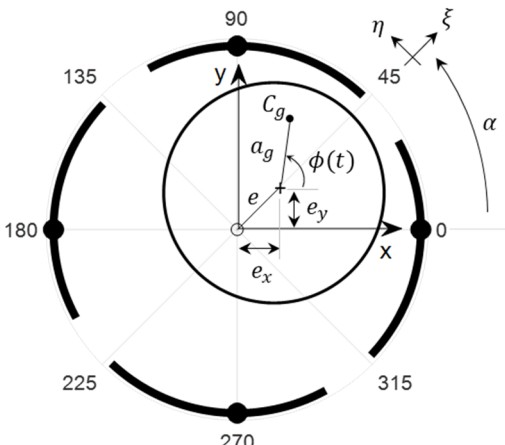

**Figure 3.** Schematic representation of the four-shoes TPJB. The global and local coordinates are given in the x-y and $\xi$-$\eta$ directions, respectively.

**Table 2.** Technical specification of the four-shoes TPJB.

|  | Descriptions | Values |
|---|---|---|
| Bearing Geometry | Number of pads | 4 |
|  | Journal diameter (mm) | 49.84 |
|  | Pad length (mm) | 20 |
|  | Pad angle (degree) | 72 |
|  | Angular pivot position (degree) | 0°, 90°, 180°, and 270° |
|  | Radial bearing clearance (mm) | 0.13 |
|  | Radial pad clearance (mm) | 0.159 |
|  | Pad pivot offset ratio (-) | 0.6 |
|  | Preload ratio (-) | 0.18 |
|  | Pad thickness (mm) | 8 |
| Material | Bearing surface material (Babbitt) |  |
|  | Thickness (mm) | 1 |
|  | Density (kg/m$^3$) | 7280 |
|  | Base pad material (Steel) |  |
|  | Thickness (mm) | 7 |
|  | Density (kg/m$^3$) | 7850 |
| Lubricant | Q6 Handel oil |  |
|  | Oil supply pressure (MPa) | 0.01 |
|  | Average inlet and outlet lubrication temperature (°C) | See Figure 4 |
|  | Viscosity at 40 °C (mPa·s) | 27.64 |
|  | Viscosity at 100 °C (mPa·s) | 6.493 |
|  | Density (kg/m$^3$) | 872 |

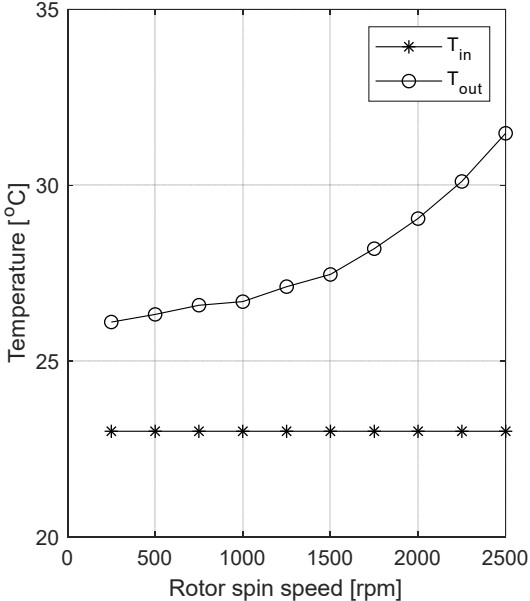

**Figure 4.** The measured inlet ($T_{in}$) and outlet ($T_{out}$) average temperatures of the lubricant are plotted as a function of the rotor speed.

## 3. Numerical Model

The numerical simulation model consists of two main parts. For the first part (described in Section 3.1), the bearing coefficients were numerically calculated using the commercial software package RAPPID [36] for a number of relative eccentricities and rotor speeds. The set of the predicted bearing coefficients was then fitted by a two-dimensional polynomial regression, and the stiffness and damping coefficient represented by polynomial equations. In the second part of the simulation procedure (described in Section 3.2), the equation of motion of the rotor rig was numerically solved using MATLAB software.

The bearing coefficients were calculated using the polynomial equations derived in the first part of the simulation procedure. Note that the two parts were carried out separately. The simulation procedure is shown, using a graphical flowchart, in Figure 5a. In addition to this, the flowchart of a classical method (Model II) is shown in Figure 5b. The bearing forces were calculated by solving the fluid film model and integrating the pressure distribution over the fluid domain at each time step. For simplification, the pressure distribution of the fluid film lubrication model was calculated by solving Reynolds equation. The simulation results from Model II are presented in Section 4.1.3. The computational efficiency of Model II was compared with that of the proposed model (Model I) based on the computational time.

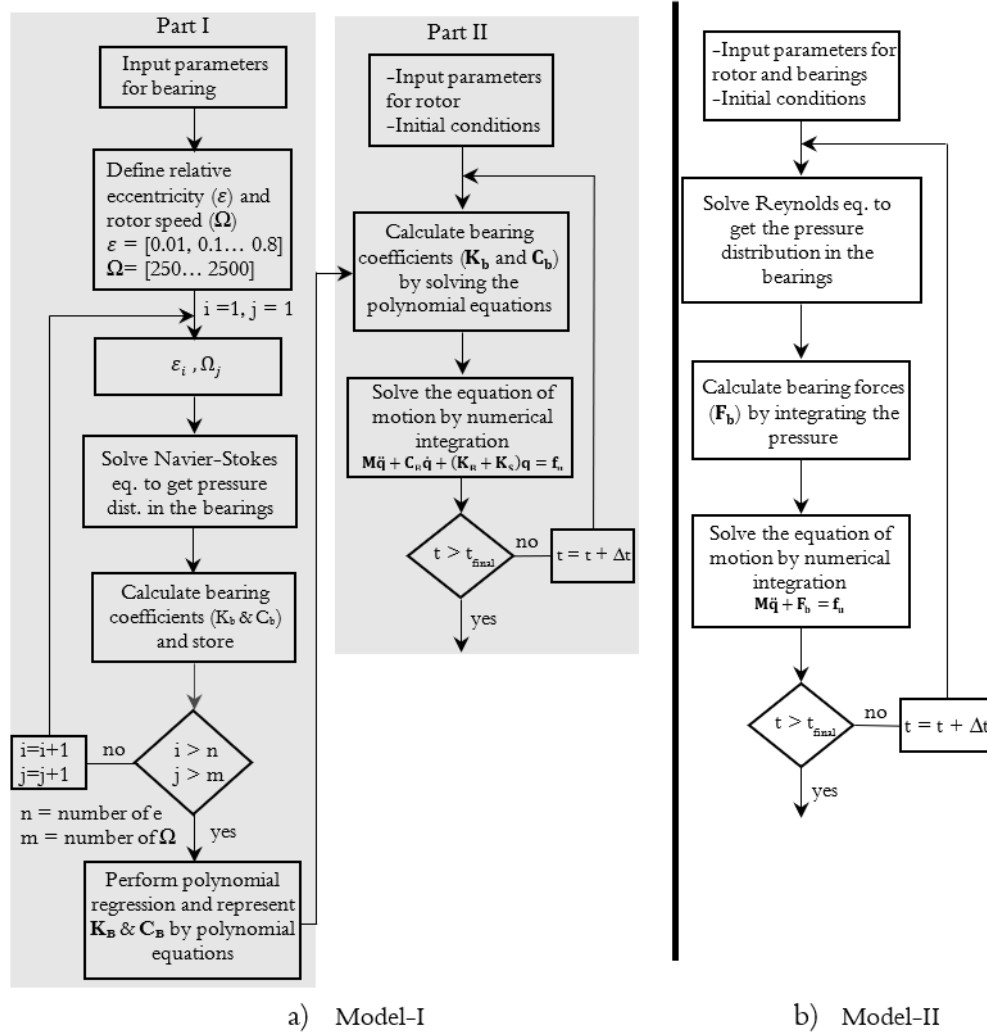

**Figure 5.** The flow chart of the (**a**) Model I: model described in this paper and (**b**) Model II: classical model.

### 3.1. Bearing Model

#### 3.1.1. Bearing Coefficients Using Fluid Film Lubrication Theory

The linearized stiffness and damping coefficients of the four-shoe TPJB were numerically computed using a commercial software package RAPPID [36] that solves the averaged Navier–Stokes based governing equations, i.e., conservation of mass, momentum, and energy. The bearing coefficients were calculated at given eccentricities, assuming no whirling of the journal. A quasi-static approximation was applied, and the whirling effect was not considered when solving the fluid-film lubrication model. In other words, a journal's tangential velocity ($V_\eta$) was assumed to be dependent only on the spinning speed of the journal. For small orbit amplitudes (e), the journal's tangential velocity due to whirling

effect can be ignored. For large orbit amplitudes (compared to the radius of the journal), however, the velocity component due to the moving journal center cannot be ignored, and quasi-static approximation could be inadequate. In this paper, the amplitude of the whirl was considerably smaller than the radius of the rotor, and the dynamic effect of the moving journal center was neglected. Thus, the quasi-static approximation is reasonable. Similarly, for large hydropower rotors, the radius ratio (e/R) is even significantly small, and the quasi-static approximation can be adequate.

All bearing components were assumed to be rigid, and their thermal and mechanical deformations were ignored. No friction between the surface of the pads and the bearing housing was considered. Besides, the mass properties of the pad were included in the dynamic solution that determines the transfer function. The pad tilts from its neutral position by rolling. This means the pivot contact point translates from its neutral position without sliding. In the neutral position, the four pivots were located at 0°, 90°, 180°, and 270°. Axial grooves between two consecutive pads were continuously fed by the supply lubricant and mixed with hot oil carried over from the preceding pad. For all simulations, the pressure and temperature of the supplied lubricant were 0.01 MPa and 23 °C, respectively. Table 3 shows the maximum temperature of the lubricant at each pad for different rotor speeds and relative eccentricity equal to 0.6. The viscosity of the lubricant was calculated by the program, which uses Vogel's Law to curve fit the data. The pressure and temperature of the lubricant at the side edges of the pads were assumed to be equal to 0.001 MPa and 23 °C. Besides, cavitation was assumed to occur when the fluid film pressure was less than or equal to 0 Pa.

**Table 3.** Maximum lubricant temperature for relative eccentricity equal to 0.6 and different rotor speeds.

| Rotor Spin Speed (RPM) | Temperature (°C) | | | |
| --- | --- | --- | --- | --- |
| | **Pad 1** | **Pad 2** | **Pad 3** | **Pad 4** |
| 250 | 26.11 | 25.91 | 25.75 | 26.05 |
| 500 | 26.32 | 25.87 | 25.55 | 26.1 |
| 1000 | 27.88 | 26.76 | 26.18 | 27.18 |
| 1500 | 29.53 | 27.58 | 26.89 | 28.35 |
| 2000 | 31.23 | 28.47 | 27.39 | 29.26 |
| 2500 | 34.28 | 30.32 | 29.16 | 31.73 |

Six different rotor speeds between 250 rpm and 2500 rpm (250 rpm, 500 rpm, 1000 rpm ... 2500 rpm), and nine different relative eccentricities (0.01, 0.1, 0.2 ... 0.8) were considered. For each rotor speed, a relative eccentricity was predefined, and the bearing coefficients were calculated for 100 different angular positions of the journal (eccentricity angle) between −45° and +45°. In Figures 6 and 7, the grey circles show the stiffness and damping coefficients that are calculated using RAPPID for 0.5 relative eccentricity at 1500 rpm rotor speed. Since the pads were assumed to be identical, the bearing coefficients obtained from the computation were periodical. Thus, they could be represented mathematically by sine and cosine functions [15] once the maximum and minimum coefficients are known. For a given relative eccentricity ($\varepsilon$) and eccentricity angle ($\alpha$), the direct and cross-coupling bearing coefficients can be expressed by Equations (1) and (2),

$$K_{ij}(\varepsilon, \alpha, \Omega) = \frac{K_{ij}^{max}(\varepsilon, \Omega) + K_{ij}^{min}(\varepsilon, \Omega)}{2} \; sign \; \frac{K_{ij}^{max}(\varepsilon, \Omega) - K_{ij}^{min}(\varepsilon, \Omega)}{2} \cdot \gamma \qquad (1)$$

$$C_{ij}(\varepsilon, \alpha, \Omega) = \frac{C_{ij}^{max}(\varepsilon, \Omega) + C_{ij}^{min}(\varepsilon, \Omega)}{2} sign \frac{C_{ij}^{max}(\varepsilon, \Omega) - C_{ij}^{min}(\varepsilon, \Omega)}{2} \cdot \gamma \qquad (2)$$

where

$$sign = \begin{cases} + & ij = \xi\xi \\ - & otherwise \end{cases}$$

$$\gamma = \begin{cases} \cos{(n\alpha)} & i = j \\ \sin{(n\alpha)} & i \neq j \end{cases}, \quad n : number\ of\ pads$$

and $K_{ij}^{max}$, $C_{ij}^{max}$, $K_{ij}^{min}$, $C_{ij}^{min}$ are the maximum and minimum bearing coefficients obtained from simulation of the fluid film lubrication model at a given relative eccentricity ($\varepsilon$) and rotor speed ($\Omega$) in the local ($i,j \rightarrow \xi,\eta$) coordinates.

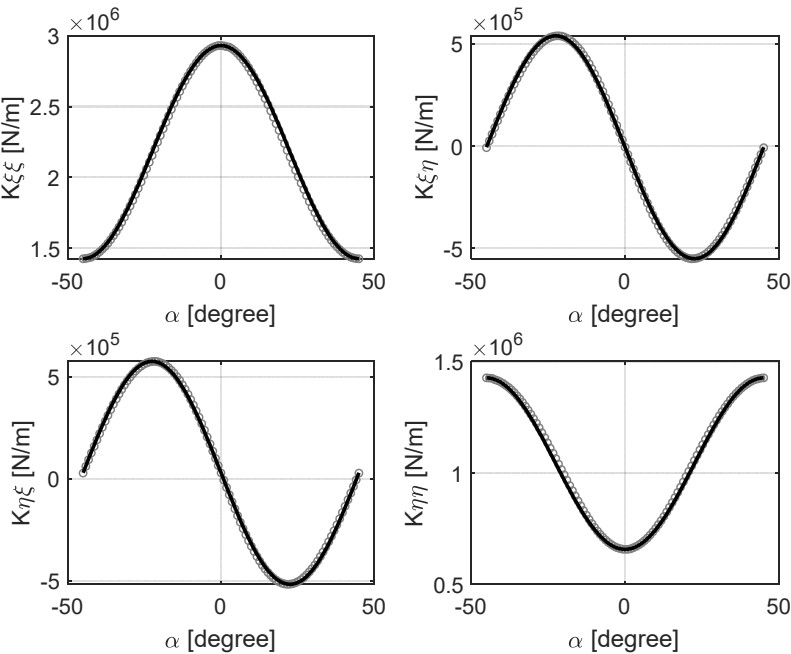

**Figure 6.** The local stiffness coefficients of the bearing at 0.5 of relative eccentricity and 1500 rpm. Circles (o) represent the stiffness coefficients calculated by RAPPID, whereas solid lines represent the stiffness coefficients calculated according to Equation (1).

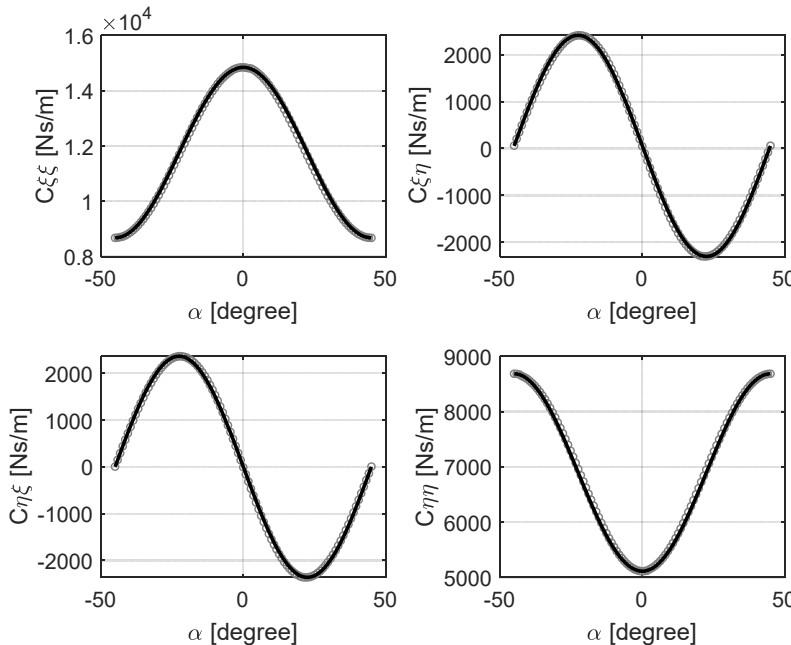

**Figure 7.** The local damping coefficients of the bearing at 0.5 of relative eccentricity and 1500 rpm. Circles (o) represent the damping coefficients calculated by RAPPID, whereas solid lines represent the damping coefficients calculated according to Equation (2).

The bearing coefficients ($\mathbf{K_\beta}$ and $\mathbf{C_\beta}$) are given in the local coordinates and have to be transformed to Cartesian coordinates prior to using them in Equation (15). Transformation of bearing matrixes was performed using Equations (3) and (4),

$$\mathbf{K_B} = \mathbf{T^T K_\beta T} \tag{3}$$

$$\mathbf{C_B} = \mathbf{T^T C_\beta T} \tag{4}$$

where $\mathbf{T}$ is a transformation matrix.

$$\mathbf{T} = \begin{bmatrix} \cos(\alpha) & \sin(\alpha) \\ -\sin(\alpha) & \cos(\alpha) \end{bmatrix} \tag{5}$$

3.1.2. Least-Square Approximation and Measures of Fitness

As discussed in the previous subsection, a total of 36 fluid film simulations were carried out, and the maximum and minimum values of each direct and cross-coupling bearing coefficients extracted. In Figures 8–11, the maximum and minimum values from the fluid film simulations are represented by dots. The approximated bearing coefficients are modeled using a two-dimensional polynomial equation as shown in Equation (6),

$$\widetilde{Y} = \sum_{i=0}^{n} \sum_{j=0}^{m} \beta_{ij} \cdot \varepsilon^i \cdot \Omega^j, \quad i+j \leq \max(r,s) \tag{6}$$

where $\widetilde{Y}$. represents an approximated local bearing coefficient and is defined as a function of relative eccentricity ($\varepsilon$) and rotor speed ($\Omega$). The degrees of relative eccentricity and ($\Omega$). The degrees of relative eccentricity and rotor speed are represented by $r$ and $s$, respectively. Regression coefficients ($\beta_{ij}$) that minimize the sum of the square error ($L$) were estimated by fitting the model using a linear least-square method in MATLAB [37]. This means that the derivatives of Equation (7) with respect to each polynomial coefficient are approximated to be zero, Equation (8).

$$L = \sum_{k=1}^{36} (Error)_k^2 = \sum_{k=1}^{36} \left( Y_k - \widetilde{Y}_k \right)^2 \tag{7}$$

$$\left. \frac{\partial L}{\partial \beta_{ij}} \right|_{\beta_{00}, \ \beta_{10}, \ \dots} = -2 \sum_{z=1}^{36} \left( Y_k - \widetilde{Y}_k \right) \cdot \varepsilon^i \cdot \Omega^j \approx 0 \tag{8}$$

The model was tested for different polynomial orders and the goodness-of-fit of each model was evaluated. A percentage relative error ($RMSE\%$) and adjusted R-square ($R_{adj}^2$) were used for evaluation. The percentage relative error was calculated by dividing the absolute error estimation ($RMSE$) with the root mean square of the response ($RMSY$) and multiplying by 100 [37], Equation (9). A low value indicates a better model. The $R_{adj}^2$ is an extension of the ordinary $R^2$ that indicates how well the fit accounts for the variation of data. It accounts for the residual degrees of freedom ($N_r$-$p$) and does not increase due to variables added to the model, which is the case for $R^2$. The values of $R_{adj}^2$ can be less than or equal to one including negative values, and the fit is better as $R_{adj}^2$ approaches one,

$$RMSE\% = \frac{RMSE}{RMSY} \cdot 100\% \tag{9}$$

$$R_{adj}^2 = 1 - \frac{SSE}{SST} \cdot \left( \frac{N_r - 1}{N_r - p} \right) \tag{10}$$

where $N_r$ and $p$ are the number of response values and polynomial coefficients, respectively.

$$SSE = \sum_{k=1}^{36} \left(Y_k - \widetilde{Y}_k\right)^2 \tag{11}$$

$$SST = \sum_{k=1}^{36} (Y_k - Y_{mean})^2 \tag{12}$$

$$RMSE = \sqrt{\frac{SSE}{N - p}} \tag{13}$$

The $RMSE\%$ and $R^2_{adj}$ of ten different models are given in Appendix A. Five different degrees of $\varepsilon^i$ ($i = 1 \dots 5$) and three degrees of $\Omega^j$ ($j = 1, 2,$ and $3$) were considered. Both $RMSE\%$ and $R^2_{adj}$ indicate that the accuracy of the fit of each coefficient increased as $n$ increased. The polynomial model with $n = 5$ and $m = 3$ showed lower values of $RMSE\%$, and $R^2_{adj}$ values that were close to one. Therefore, *poly53* (a MATLAB inbuilt function) was chosen as it was relatively the best fit model compared to the other models presented in this paper. All values of $RMSE\%$ and $R^2_{adj}$ of the *poly53* fit were <10% and >0.99, respectively. The bearing coefficients were then fitted using *poly53* as shown in Figures 8–11, and the regression coefficients of Equation (14) are given in Appendix B.

$$\begin{aligned}
\widetilde{Y} = {}& \beta_{00} + \beta_{10}{\cdot}\varepsilon + \beta_{01}{\cdot}\Omega + \beta_{20}{\cdot}\varepsilon^2 + \beta_{11}{\cdot}\varepsilon{\cdot}\Omega + \beta_{02}{\cdot}\Omega^2 + \beta_{30}{\cdot}\varepsilon^3 + \beta_{21}{\cdot}\varepsilon^2{\cdot}\Omega \\
&+ \beta_{12}{\cdot}\varepsilon{\cdot}\Omega^2 + \beta_{03}{\cdot}\Omega^3 + \beta_{40}{\cdot}\varepsilon^4 + \beta_{31}{\cdot}\varepsilon^3{\cdot}\Omega + \beta_{22}{\cdot}\varepsilon^2{\cdot}\Omega^2 \\
&+ \beta_{13}{\cdot}\varepsilon{\cdot}\Omega^3 + \beta_{50}{\cdot}\varepsilon^5 + \beta_{41}{\cdot}\varepsilon^4{\cdot}\Omega + \beta_{32}{\cdot}\varepsilon^3{\cdot}\Omega^2 + \beta_{23}{\cdot}\varepsilon^2{\cdot}\Omega^3
\end{aligned} \tag{14}$$

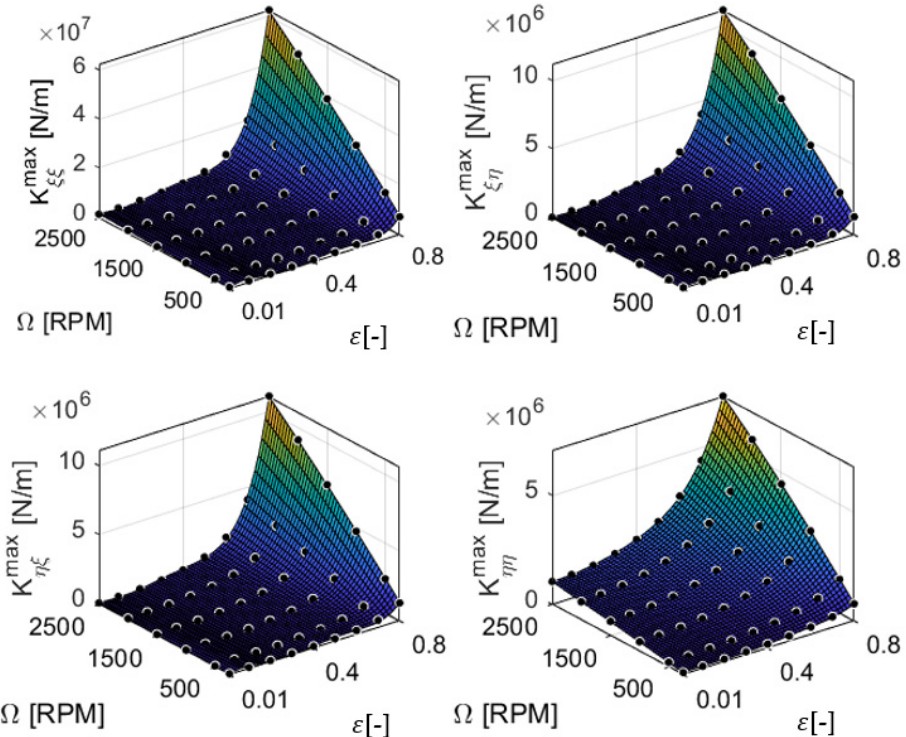

**Figure 8.** The maximum stiffness coefficients in the local coordinates, fitted using MATLAB (*poly53*). The dots represent the results from the fluid film simulations.

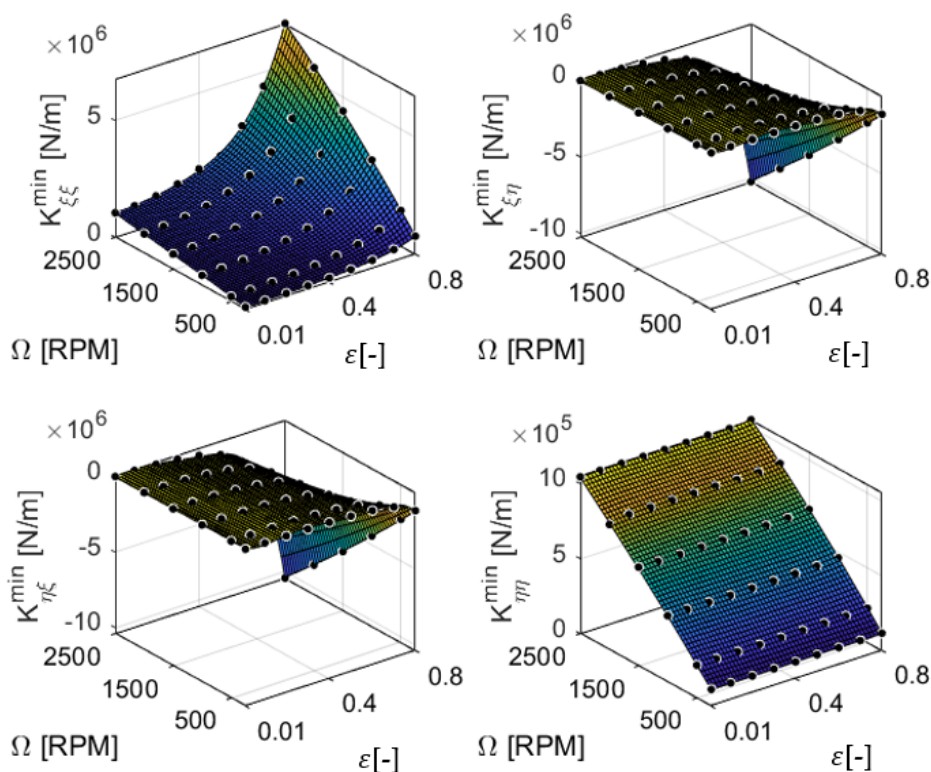

**Figure 9.** The minimum stiffness coefficients in the local coordinates, fitted using MATLAB (*poly53*). The dots represent the results from the fluid film simulations.

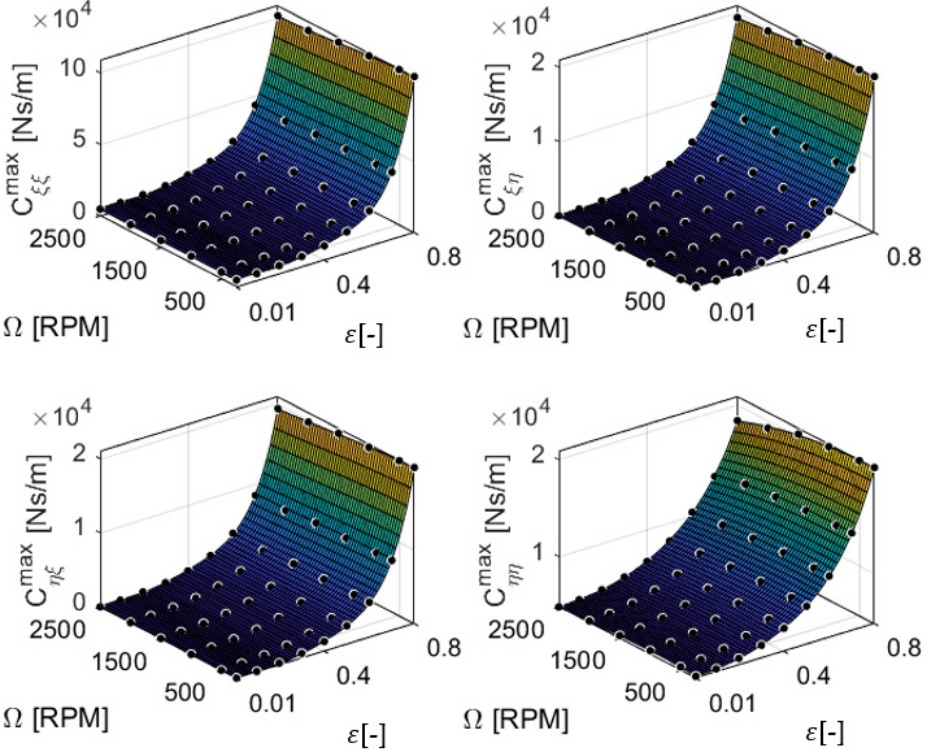

**Figure 10.** The maximum damping coefficients in the local coordinates, fitted using MATLAB (*poly53*). The dots represent the results from the fluid film simulations.

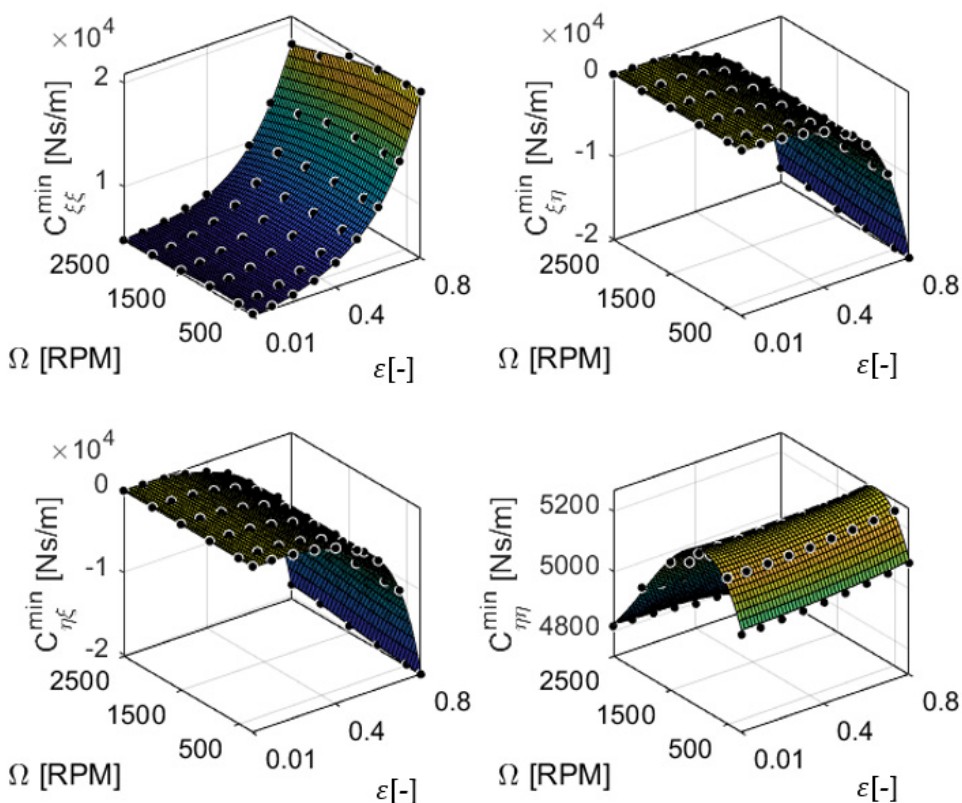

**Figure 11.** The minimum damping coefficients in the local coordinates, fitted using MATLAB (*poly53*). The dots represent the results from the fluid film simulations.

### 3.2. Rotor Rig Model

In the second part of the numerical simulation procedure, the unbalance response of the rotor rig was simulated. Figure 2 shows the schematic representation of the rotor rig, which comprises the rotor and supports. The first natural frequency of the rotor is higher than the frequency region of interest, and it is, therefore, modeled as a rigid rotor. The numerical model of the rotor rig includes the rotor and two identical supporting structures, representing a total of eight degrees of freedom (DOF), $\mathbf{q} = [x_1\ y_1\ x_2\ y_2\ x\ y\ \theta\ \psi]^T$. In the vector $\mathbf{q}$, the first four variables are the DOF of the upper and lower supports, whereas the last four variables are the DOF of the rotor. Each support is composed of the four-shoes TPJB and the bracket.

The rotor was modeled as a rigid rotor and represented by a mass matrix ($\mathbf{M}$) and gyroscopic matrix ($\mathbf{G}$). Equation (15) show the general equation of motion, and it was rearranged in the state-space formulation and solved numerically using MATLAB (*ode15s*). The brackets had a direct stiffness of 500 MN/m, represented by $\mathbf{K_S}$. Depending on the speed and journal location, the stiffness ($\mathbf{K_B}$) and damping ($\mathbf{C_B}$) coefficients of the bearings varied.

$$\mathbf{M}\ddot{\mathbf{q}} + \left(\dot{\phi}\mathbf{G} + \mathbf{C_B}\right)\dot{\mathbf{q}} + \left(\ddot{\phi}\mathbf{G} + \mathbf{K_B} + \mathbf{K_S}\right)\mathbf{q} = \mathbf{f_u} \tag{15}$$

where is the angular speed, $\ddot{\phi}$ is the angular acceleration, $\mathbf{f_u}$ is the unbalance force vector.

$$\mathbf{f_u} = \begin{Bmatrix} m_u d\dot{\phi}^2 \cos(\phi) + m_u d\ddot{\phi} \sin(\phi) \\ m_u d\dot{\phi}^2 \sin(\phi) - m_u d\ddot{\phi} \cos(\phi) \\ 0 \\ 0 \end{Bmatrix} \tag{16}$$

For the response simulations under constant rotor spin speed ($\dot{\phi} = \Omega$), the angular acceleration is zero ($\ddot{\phi} = 0$). The simulations were carried out for 200 shaft revolutions and responses at the beginning of the simulation, which occurred before the rotor reached its stable closed-orbit response, were ignored. The unbalance force ($\mathbf{f_u}$), which is due to the unbalance mass ($m_u$) located at a distance $d$ from the axis of the rotor, has a nonlinear relationship with the speed of the rotor ($\Omega$), Equation (16).

Unlike the response simulations under constant rotor spin speed, the angular velocity of the rotor ($\dot{\phi}$) in the transient response simulations changes linearly with time. Three different unbalance magnitudes and three different angular acceleration rates were considered. In the simulations, the ramp-up of the rotor speed started from 250 rpm and linearly increased to 2500 rpm. Low-speed operations (<250 rpm) were not considered in the simulation as the bearings operated at low eccentricities and displayed strong nonlinearity.

## 4. Results

### 4.1. Responses under Constant Rotor Spin Speed

#### 4.1.1. Orbits and Bearing Reaction Forces

The simulated shaft displacements and bearing reaction forces were investigated and compared with the experiments. Both the upper and lower four-shoe TPJBs were analyzed. Figure 12 shows the responses from the simulation and experimental results for an unbalance magnitude of $5.9 \times 10^{-3}$ kg·m and 500 rpm, 1500 rpm, and 2500 rpm rotor speeds. All the response measurements have been filtered with a fourth-order Butterworth bandpass filter, using lower and upper cut-off frequencies of 3 Hz and $10 \times \Omega$, respectively. The responses from the original measurements, so the unfiltered data (UF), have been plotted together with the responses of the filtered data (F) and the simulation results (S). Besides, the average values of the filtered data, denoted by Fm, are shown in the same figure. The simulation results (S) represent the displacement and force responses calculated for 200 shaft revolutions. However, the results from the beginning of the simulations, before they reached stable closed orbits, have been excluded from the analysis.

The shapes of the simulated orbits and forces are similar to those from the experiments. Both the orbits and forces are square-shaped since the bearing coefficients vary depending on the position of the shaft. The stiffness and damping coefficients are relatively larger when the journal is located on pads, resulting in lower shaft displacement and larger bearing forces. The orbits and bearing forces for $1.7 \times 10^{-3}$ kg·m and $3.8 \times 10^{-3}$ kg·m are shown in Appendix C. Similarly, the shapes of the orbits and forces are square, though they are not as amplified as the orbits for larger unbalance magnitudes. At lower unbalance magnitudes and rotor speeds, the amplitudes of the displacements and bearing forces are relatively lower, and the orbits and forces are round.

#### 4.1.2. Resultant Displacements and Forces

The resultant displacements ($\sqrt{e_x^2 + e_y^2}$) and forces ($\sqrt{F_x^2 + F_y^2}$) were calculated for both simulations (S) and experiments (Fm). Figures 13 and 14 show the summary of the results for different unbalance magnitudes and rotor speeds. Data points were fitted in MATLAB using piecewise cubic interpolation for visualization. An amplitude peak close to 1000 rpm was observed in the measurements due to the structural resonance of the Plexiglas shield. This amplitude peak did not exist in the simulations. The results of the simulations and experiments show similar trends, and the amplitude of the responses increases with the unbalance magnitude and rotor speed. For both upper and lower bearings, the maximum resultant displacements and forces from the numerical simulations were 90.8 μm ($\varepsilon = 0.69$) and 360.3 N. Similarly, in the experiments, the shaft was displaced by a maximum of 89.3 μm ($\varepsilon = 0.68$) and 100.5 μm ($\varepsilon = 0.77$) at the upper and lower bearings, respectively. The maximum force acting on the upper and lower bearings was 416.5 N and 403 N, respectively.

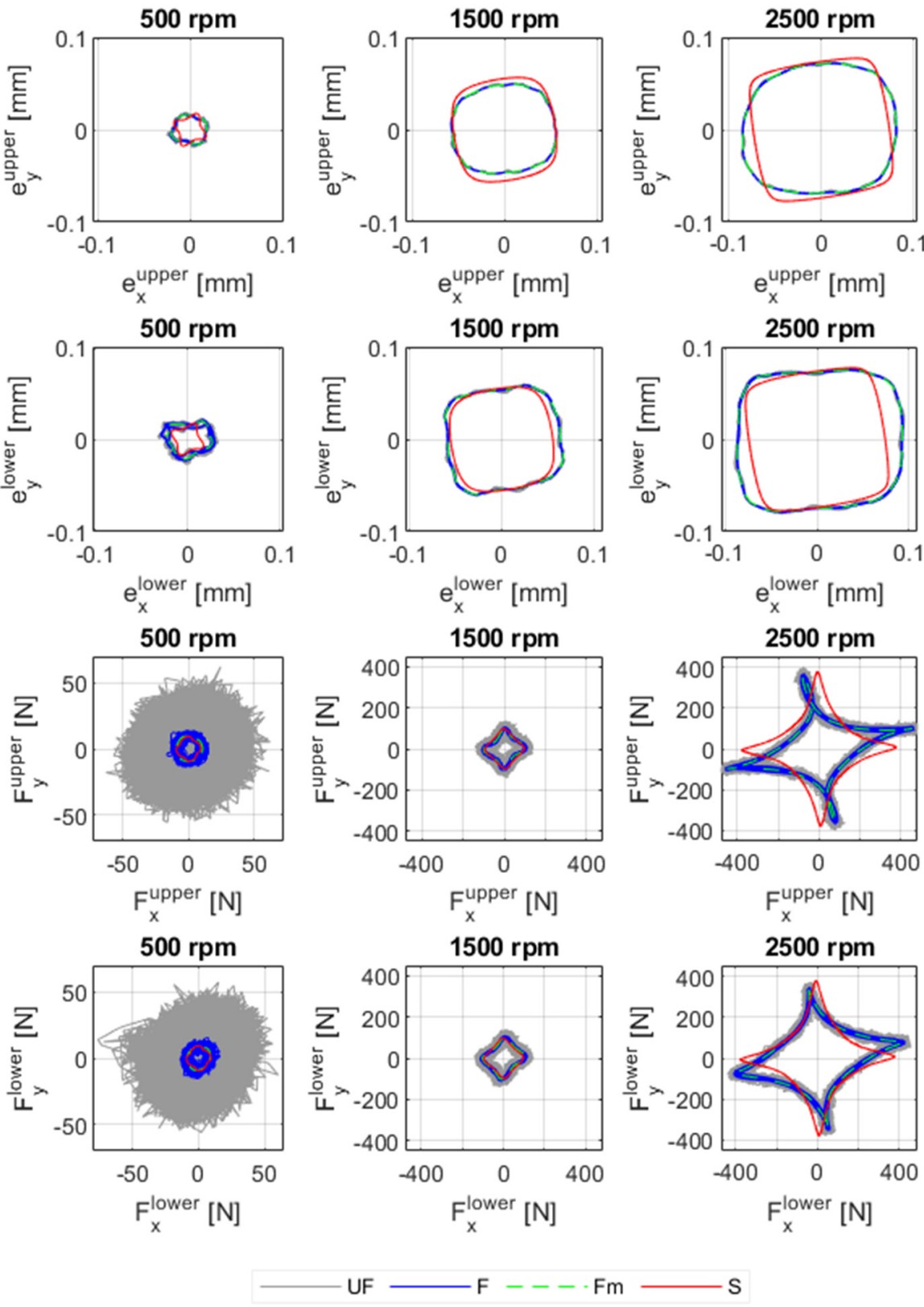

**Figure 12.** Orbits and bearing forces for the upper and lower bearings with $5.9 \times 10^{-3}$ kg·m unbalance magnitude and at 500 rpm, 1500 rpm and 2500 rpm: UF: unfiltered measurement signal; F: filtered measurement signal by a band pass filter; Fm: the average value of the filtered measurement signal; S: simulation.

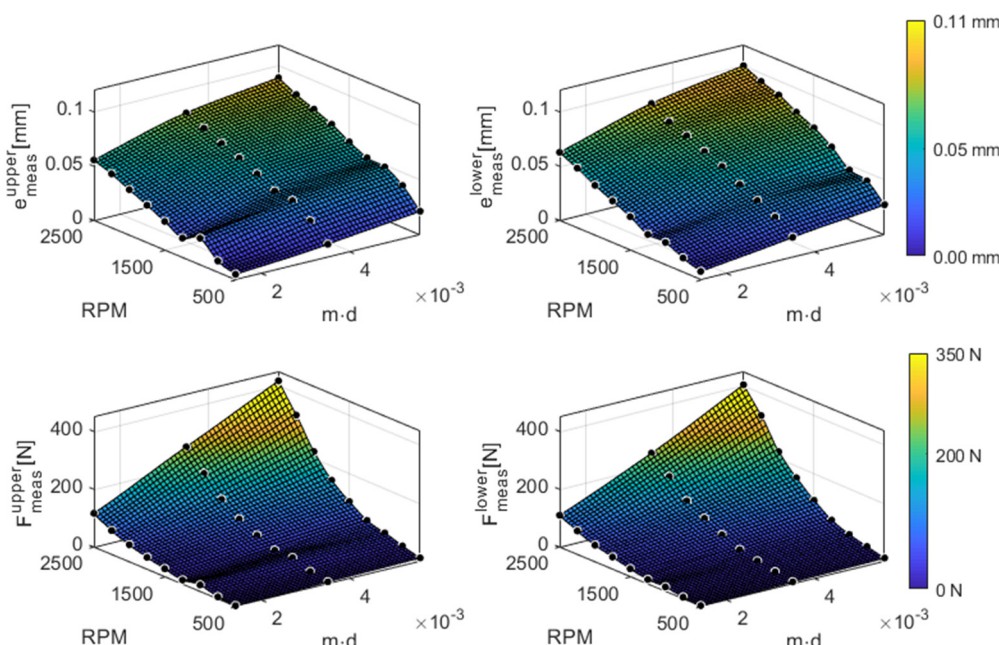

**Figure 13.** Summary of the measurements showing the mean values of the resultant displacement and reaction forces for the upper and lower bearings. The data points are shown with black circles.

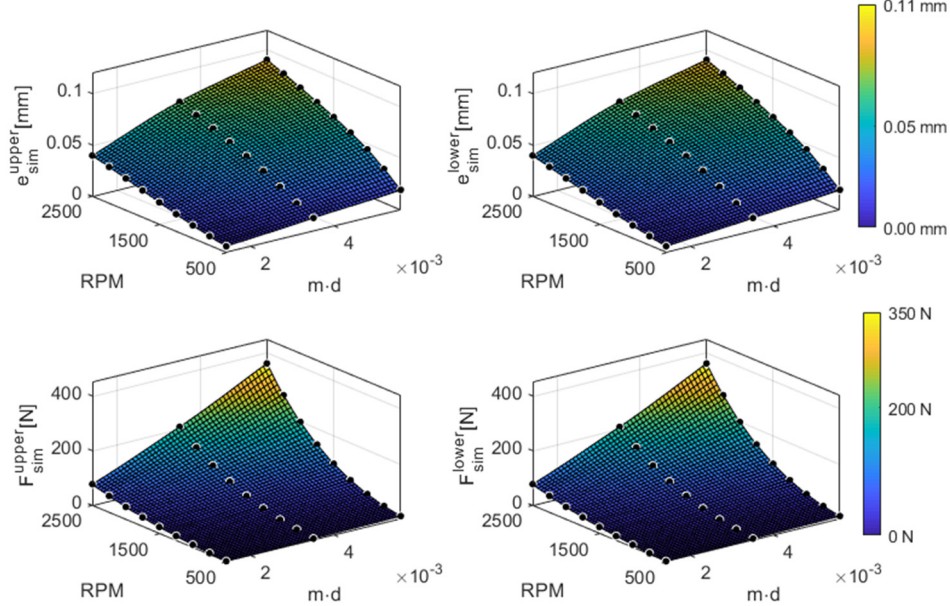

**Figure 14.** Summary of the simulation result showing the mean values of the resultant displacement and reaction forces for the upper and lower bearings. The data points are shown with black circles.

In order to evaluate the accuracy of the model, the proportional deviations of the simulations from the experiments were calculated using Equations (17) and (18). Table 4 shows the percentage of relative errors of the resultant displacement (%$E_{disp}^i$) and bearing force (%$E_{Force}^i$) at each rotor speed, with the values ranging from zero to positive infinity. A value close to zero indicates a better correlation between the simulation and the experiment. The simulation deviated by a maximum of 54.1% from the experiment, for $1.7 \times 10^{-3}$ kg·m unbalance magnitude and 1000 rpm rotor speed. This is expected since the measurements are affected by structural resonance at this rotor speed. Excluding the comparison at

1000 rpm, the maximum deviation was 39.4%. With some exceptions, the deviation was relatively higher at lower rotor speeds and lower unbalance magnitudes.

$$\%\mathrm{E}_{disp}^i = \left| \frac{e_{exp}^i - e_{sim}^i}{e_{exp}^i} \right| \times 100\% \tag{17}$$

$$\%\mathrm{E}_{Force}^i = \left| \frac{F_{exp}^i - F_{sim}^i}{F_{exp}^i} \right| \times 100\%, \quad i := upper, lower \tag{18}$$

**Table 4.** The percentage relative errors of the resultant displacement ($\%\mathrm{E}_{disp}^i$) and the bearing force ($\%\mathrm{E}_{Force}^i$).

| | $m \cdot d$ | RPM | | | | | | | | |
|---|---|---|---|---|---|---|---|---|---|---|
| | | 500 | 750 | 1000 | 1250 | 1500 | 1750 | 2000 | 2250 | 2500 |
| $\%\mathrm{E}_{disp}^{upper}$ | $1.7 \times 10^{-3}$ | 18.2 | 13 | 48 | 5.5 | 18.5 | 23.9 | 26.2 | 28.7 | 28.7 |
| | $3.8 \times 10^{-3}$ | 6.9 | 24 | 23.3 | 6 | 7.4 | 8.3 | 7.8 | 8.3 | 9 |
| | $5.9 \times 10^{-3}$ | 10.2 | 14.6 | 7.6 | 8.6 | 8.2 | 4.5 | 4.4 | 4.4 | 1.6 |
| $\%\mathrm{E}_{disp}^{lower}$ | $1.7 \times 10^{-3}$ | 20.8 | 30.7 | 38.2 | 20.5 | 30 | 31.7 | 32.6 | 35.5 | 36.7 |
| | $3.8 \times 10^{-3}$ | 29.9 | 32.7 | 23.5 | 18 | 18.7 | 17.2 | 17 | 15.2 | 18 |
| | $5.9 \times 10^{-3}$ | 29.9 | 23.2 | 1.8 | 9.4 | 11.6 | 10.3 | 11.3 | 9.2 | 9.6 |
| $\%\mathrm{E}_{Force}^{upper}$ | $1.7 \times 10^{-3}$ | 35.6 | 31.8 | 54.1 | 11.9 | 14.4 | 19 | 22.6 | 26.9 | 33.5 |
| | $3.8 \times 10^{-3}$ | 17.5 | 20.5 | 38.9 | 0.9 | 9.1 | 10.9 | 14.3 | 21.1 | 22.3 |
| | $5.9 \times 10^{-3}$ | 15.2 | 23.2 | 16.8 | 0.1 | 5.2 | 5.8 | 11.4 | 16.6 | 13.5 |
| $\%\mathrm{E}_{Force}^{lower}$ | $1.7 \times 10^{-3}$ | 39.4 | 21.7 | 35.8 | 0.1 | 14.3 | 21.1 | 20.9 | 25.8 | 29.5 |
| | $3.8 \times 10^{-3}$ | 13.4 | 13.1 | 14.3 | 6.3 | 12 | 13 | 10.9 | 16.9 | 15.5 |
| | $5.9 \times 10^{-3}$ | 10 | 17.4 | 5.9 | 1.8 | 8.8 | 10.9 | 11.2 | 16.2 | 10.6 |

### 4.1.3. Computational Time

In the previous two subsections, the accuracy of the simulation model presented in this paper (Model I) was evaluated. The results from the unbalance response simulations were compared with the results from the experiments. In this subsection, the computational efficiency of the model in reducing the simulation time is described. Model I was compared with the classical simulation approach (Model II), which requires solving the fluid film lubrication models at each time step. Flow charts for the two numerical models are shown in Figure 5. The unbalance response of the rotor rig for $5.9 \times 10^{-3}$ kg·m and 1500 rpm was simulated for 0.5 s (i.e., 12.5 revolutions), and the computation time required by the two models were compared. All the simulations, which are presented in this subsection, were carried out on a standard consumer laptop (Intel Core i7-8850H and CPU at 2.60 GHz) using MATLAB 2019a software.

For Model II, the unbalance response of a rotor rig was numerically simulated by employing FEM to solve the Reynolds equation. Descriptions of the Reynolds equation and the simulation procedure are given in Appendix D of this article and in [38–40]. For simplification, a plain cylindrical journal bearing with no groove was considered. As with the four-shoe TPJB, the diameter of the journal and the axial length of the plain bearing were 49.85 mm and 20 mm, respectively. The fluid film domain was discretized into the $N_e \times M_e$ number of elements, where $N_e$ is the number of elements in the circumferential direction and $M_e$ is the number of elements in the axial direction. The pressure distribution of the fluid film lubrication was numerically calculated by solving the Reynolds equation using FEM.

For Model I, the polynomial equations from the first part (Part I in Figure 5a) were used for all simulations without redoing the entire analysis. The computational time required to compute the unbalance response, Part II of the simulation procedure, was 0.897 s. The accuracy and computational time needed to run simulations using Model II are dependent on the number of elements used in the FEM. Table 5 shows the results of

different unbalance response simulations and their corresponding computational time. By increasing the number of elements, the accuracy of the simulation improves, which, in turn, increases the simulation time. For illustration, the values in Table 5 are plotted in Figure 15. As the number of elements increases, the relative eccentricity and force converge to exact solutions, and the curves approach towards flat horizontal lines. Regardless of the accuracy, Model I was at least three times faster than Model II in terms of computation time. For low numbers of elements, the accuracy of the results is noticeably poor, and the computation time increases depending on the level of accuracy required. Moreover, unlike plain cylindrical journal bearings, the TPJB model includes additional degrees of freedom due to the motion of the pads, resulting in greater computational effort. Therefore, Model II would have taken longer computational time if TPJBs were modeled instead of plain cylindrical journal bearings.

**Table 5.** The unbalance response simulation results for Model II for different number of mesh elements. A MATLAB tic-toc command was used to find the computational time.

| Mesh ($N_e \times M_e$) | Computational Time [sec] | Relative Eccentricity [-] | Force [N] |
|:---:|:---:|:---:|:---:|
| $6 \times 4$ | 3.31 | 73.91 | 69.08 |
| $8 \times 6$ | 5.43 | 63.05 | 75.75 |
| $14 \times 9$ | 10.65 | 59.05 | 79.96 |
| $18 \times 12$ | 19.08 | 58.86 | 81.34 |
| $27 \times 18$ | 40.75 | 58.69 | 82.95 |
| $36 \times 24$ | 91.37 | 59.01 | 83.36 |
| $54 \times 36$ | 297.92 | 59.17 | 84.06 |
| $90 \times 60$ | 2000.10 | 59.34 | 84.63 |

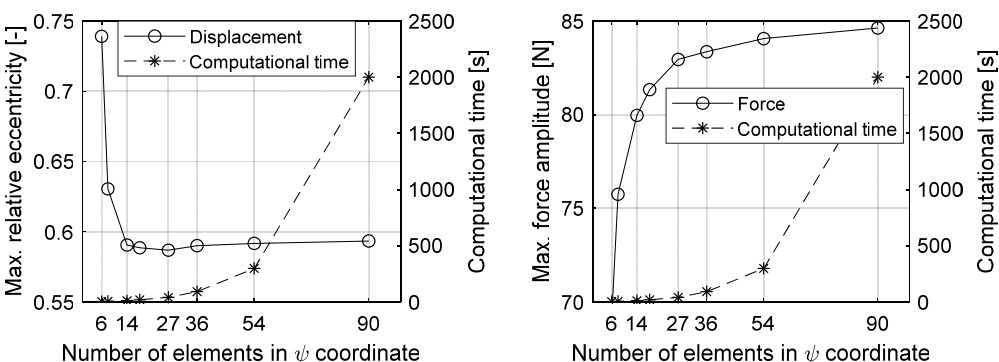

**Figure 15.** The computational time plotted together with the relative eccentricity and force amplitudes as a function of number of elements in circumferential coordinates.

Figure 16 shows the unbalance responses of the two simulation models. In fact, the two models simulate two different bearing types, so the results are expected to be different. Unlike Model I, both the orbits and bearing forces of Model II are circular since the bearings behave in similar ways regardless of the angular position of the journal. The dynamic coefficients of the four-shoe TPJB in Model I, however, vary periodically over the bearing's circumferential angle, leading to square-shaped orbits. Apart from the influence of pads in the TPJB model, however, the results from Model I agree with those from Model II, especially for finer meshes.

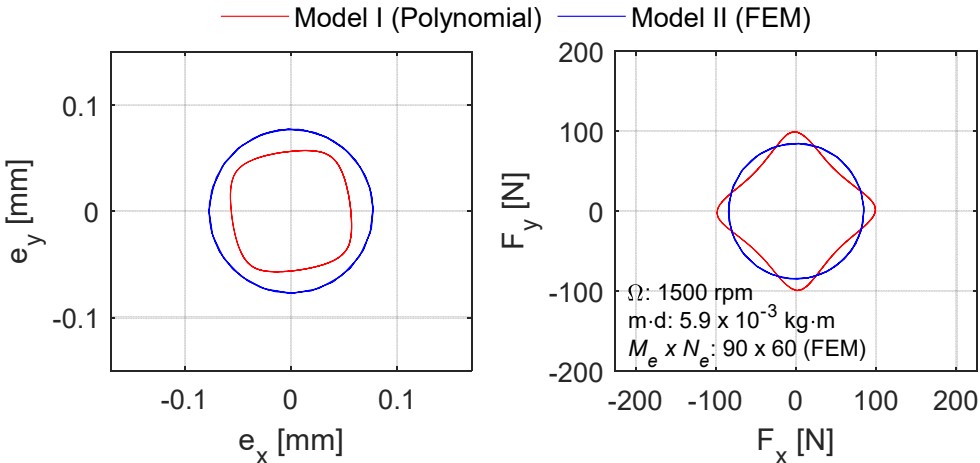

**Figure 16.** The orbits and bearing forces from the two simulation models.

## 4.2. Transient Responses

### 4.2.1. Displacements and Bearing Reaction Forces

The simulation and experimental results with different unbalance magnitudes and angular acceleration rates are presented. Figures 17 and 18 display the displacements and bearing forces of the upper and lower bearings with $5.9 \times 10^{-3}$ kg·m unbalance magnitude and $2\pi$ rad/s$^2$ angular acceleration. The responses from both simulation and experiment show a similar trend, and all the displacements and bearing forces increase with the rotor speed. However, unlike in the simulation, an amplitude peak appears close to 1000 rpm in the experiment, which is due to the structural resonance of the test rig. In the experiment, the upper and lower bearings perform in the same way, and with slight differences. Excluding responses close to 1000 rpm, the maximum deviation of displacement and force amplitude between the two bearings are 0.0122 mm (18%) and 39.2 N (10%).

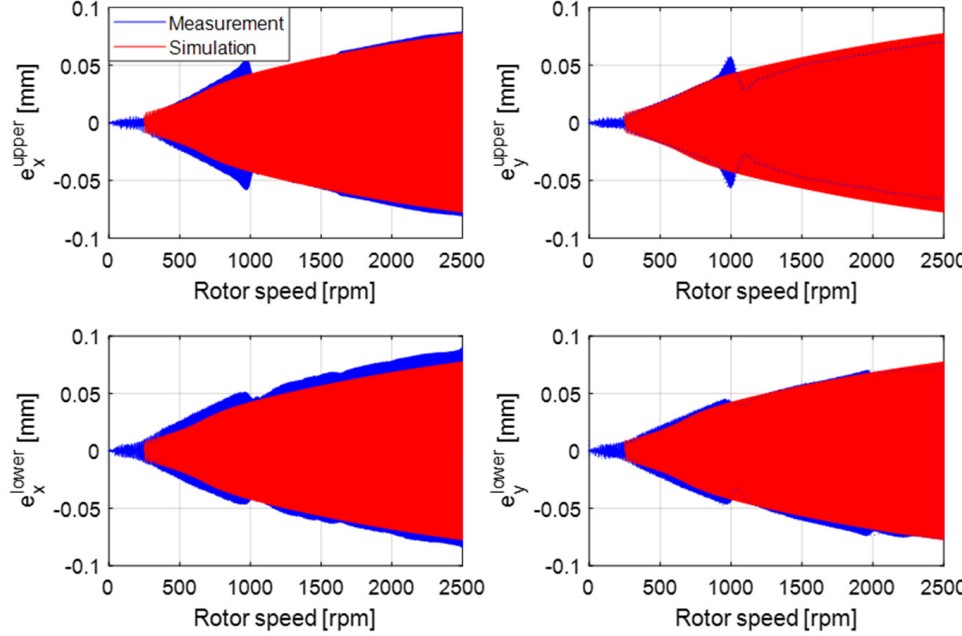

**Figure 17.** The simulated and measured displacements of the shaft with $5.9 \times 10^{-3}$ kg·m unbalance magnitude and $2\pi$ rad/s$^2$ angular acceleration.

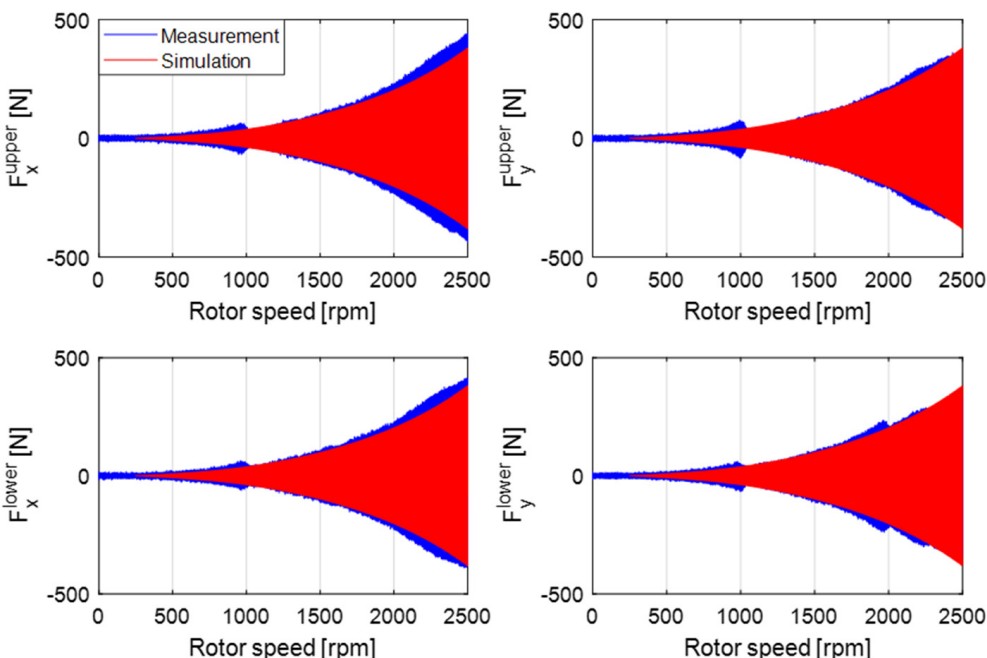

**Figure 18.** The simulated and measured bearing forces with $5.9 \times 10^{-3}$ kg·m unbalance magnitude and $2\pi$ rad/s$^2$ angular acceleration.

Figures 19 and 20 show the orbits and bearing forces for three ranges of rotor speeds, i.e., (500–600) rpm, (1250–1350) rpm, and (2250–2350) rpm. Both the simulation and experimental results show similar patterns. The periodically varying bearing coefficients on and between pads produce orbits and forces with peaks and valleys. For lower unbalance magnitude and rotor speed, however, the shape of the orbits looks relatively circular. This is because the bearings operate at low orbit amplitudes and the bearing coefficients are almost independent of the angular position of the journal.

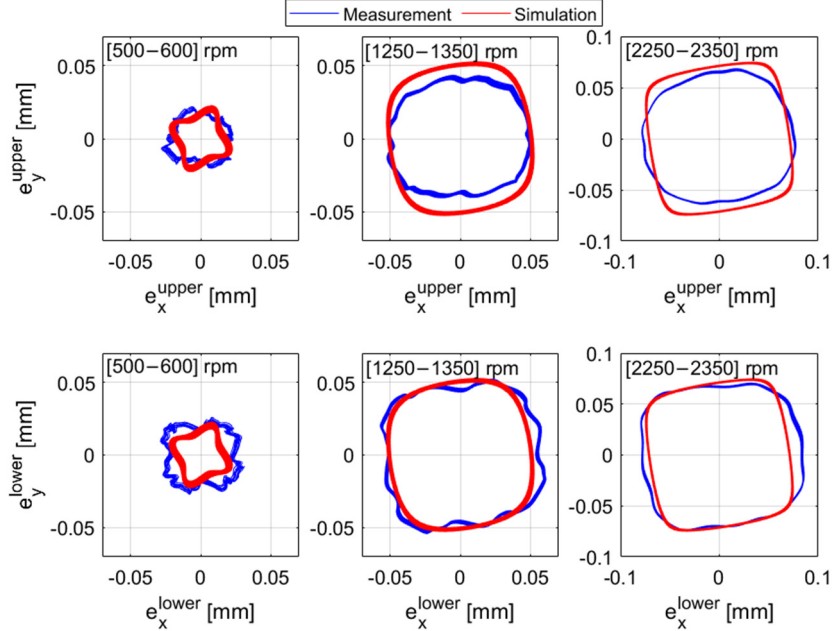

**Figure 19.** The simulated and measured orbit of the shaft with $5.9 \times 10^{-3}$ kg·m unbalance magnitude and $2\pi$ rad/s$^2$ angular acceleration.

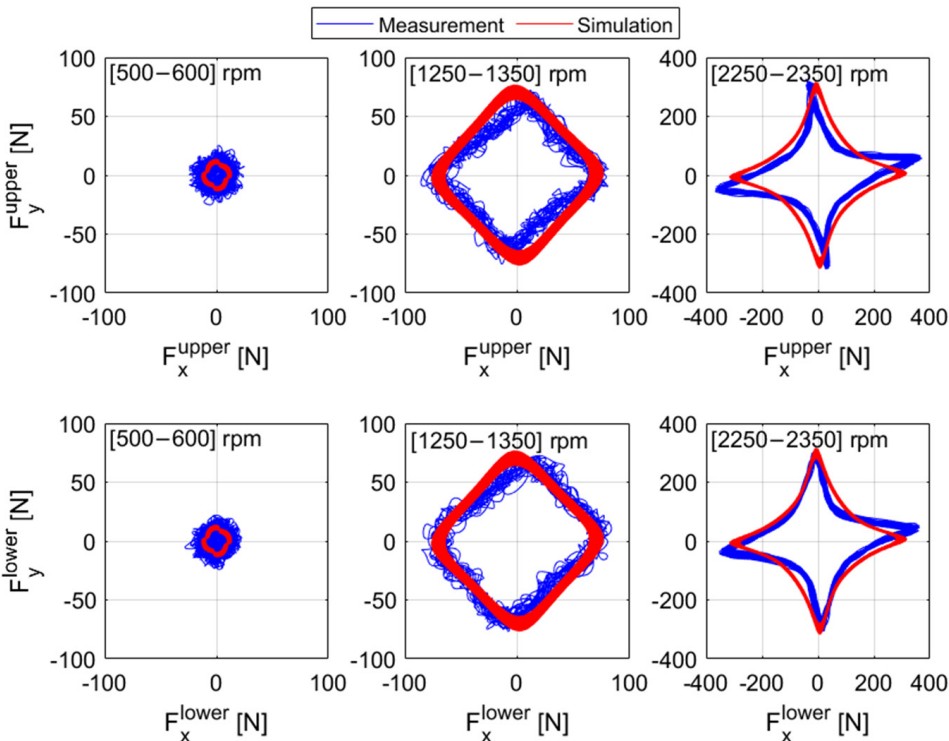

**Figure 20.** The simulated and measured bearing force with $5.9 \times 10^{-3}$ kg·m unbalance magnitude and $2\pi$ rad/s$^2$ angular acceleration.

#### 4.2.2. Resultant Displacements and Forces

Figures 21 and 22 summarize the results from the transient simulations and experiments for three different unbalance amplitudes (i.e., $1.7 \times 10^{-3}$ kg·m, $3.8 \times 10^{-3}$ kg·m and $5.9 \times 10^{-3}$ kg·m) and three different angular accelerations (i.e., $0.5\pi$ rad/s$^2$, $2\pi$ rad/s$^2$ and $4\pi$ rad/s$^2$). The response amplitudes for rotor speeds between 250 rpm and 2500 rpm were considered and divided into 23 series intervals with a block size of 100 rpm (except the last interval, whose block size is 50 rpm). In each interval, the maximum resultant responses were calculated and presented as a function of rotor speed. Both the maximum resultant displacements and bearing forces increased with the rotor speed. The results from the simulations and experiments followed a similar trend. As stated above, the structural resonance of the test rig, which is around 1000 rpm, appears in all the results from the measurements.

As shown in Figures 21 and 22, the numerical simulation predicts the actual measurements well with a slight deviation. For both displacements and forces, these deviations decrease as the unbalance magnitude increases. At $5.9 \times 10^{-3}$ kg·m unbalance magnitude, the simulation resultant displacement and force deviate from the corresponding measurement with a maximum of 10% and 14% percentage relative deviation (disregarding responses at 1000 rpm), respectively. For both simulations and experiments, the results with three different angular acceleration rates are similar and the magnitude of the responses is not significantly affected.

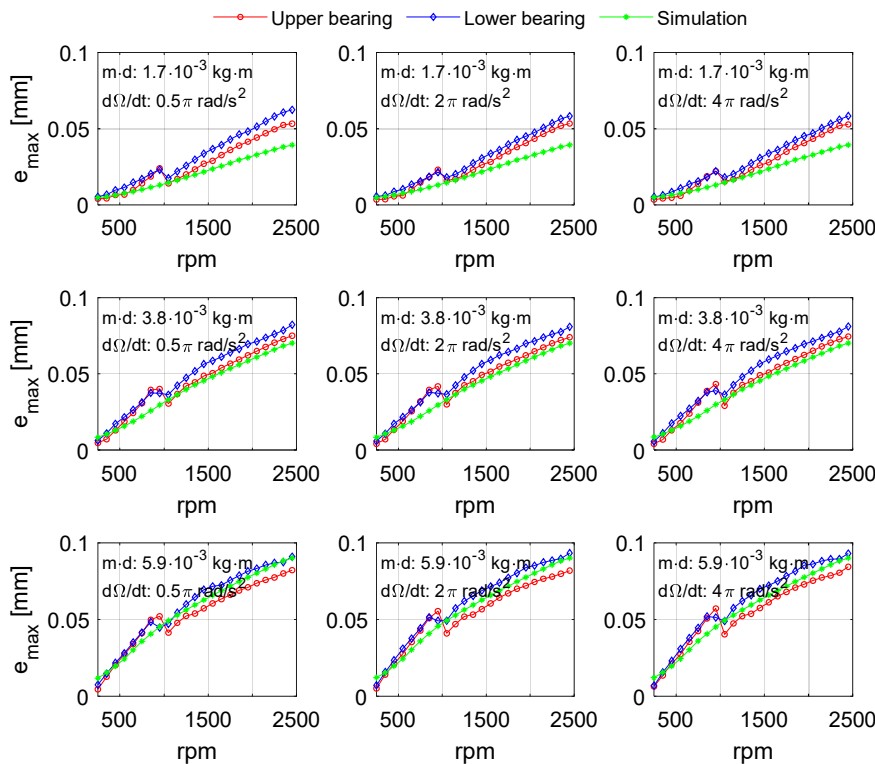

**Figure 21.** The simulated and measured shaft displacement for nine different cases: m·d: unbalance magnitude and dΩ/dt: angular acceleration.

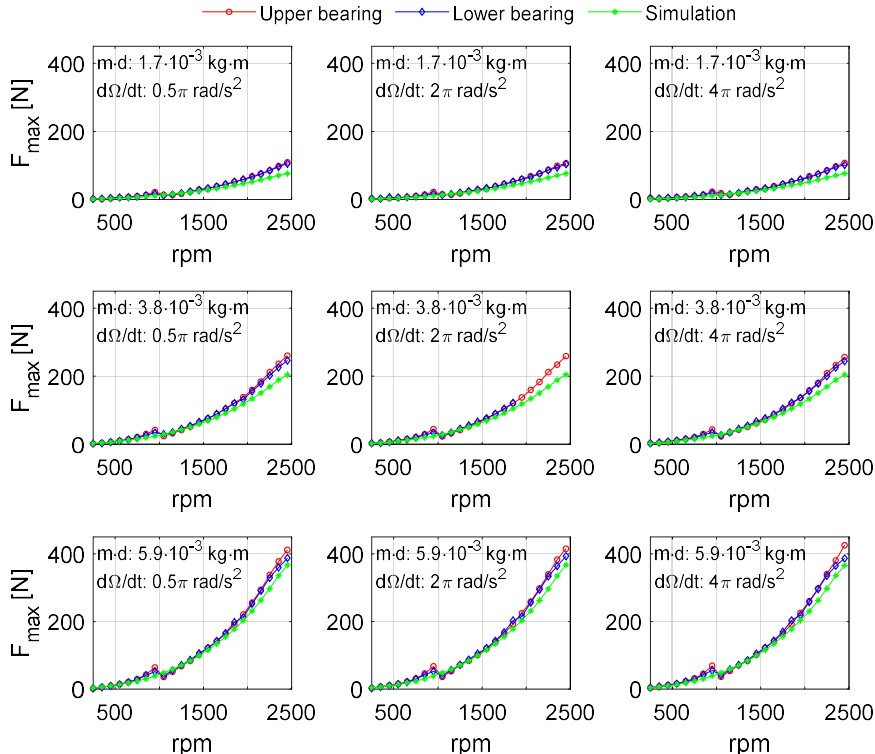

**Figure 22.** The simulated and measured bearing forces for nine different cases: m·d: unbalance magnitude and dΩ/dt: angular acceleration. The lower bearing force measurement with $3.8 \times 10^{-3}$ kg·m and $2\pi$ rad/s$^2$ has some discrepancies after 1950 rpm and is excluded from the analysis.

### 4.2.3. Frequency Response Function (FFT)

The frequency content of the responses was calculated using a fast Fourier transform (FFT) in the stationary x and y directions. Figures 23 and 24 show the waterfall diagram of the displacement responses from the simulations and experiments, respectively. Similarly, the simulated and measured bearing forces are shown in Figures 25 and 26, respectively. Results from both simulation and experiment show that the first-order frequency ($1 \times \Omega$), which is due to the unbalance mass, is the dominant frequency. Besides, amplitude peaks at the third ($3 \times \Omega$) and fifth ($5 \times \Omega$) frequency orders exist in both simulation and experiment. These frequencies are associated with the number of pads ($n = 4$) [34] and they exist as a single frequency at $4 \times \Omega$ in the rotating coordinates.

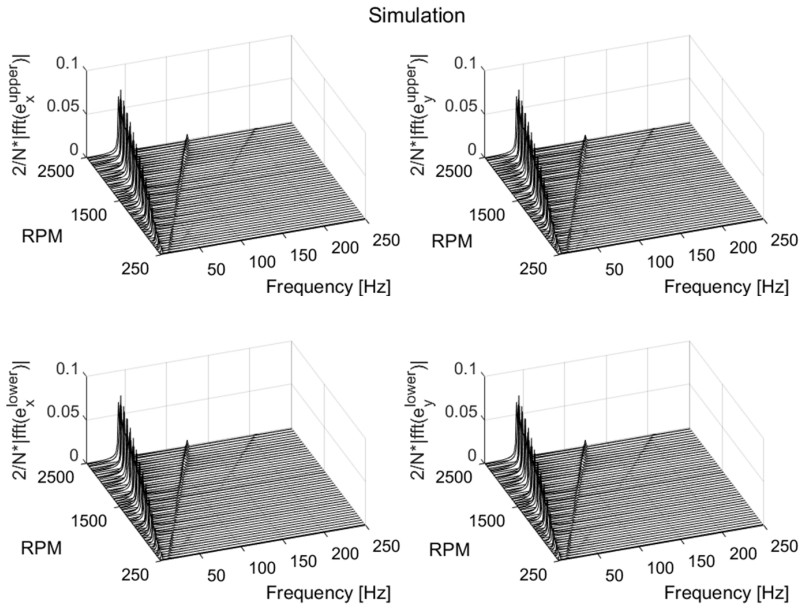

**Figure 23.** The waterfall diagram of the simulated shaft displacement with $5.9 \times 10^{-3}$ kg·m and $2\pi$ rad/s$^2$. N is the number of samples.

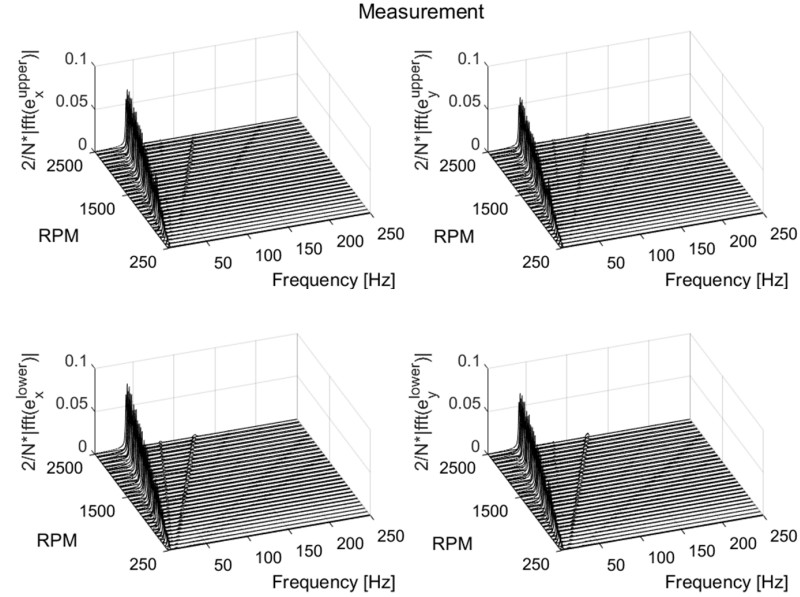

**Figure 24.** The waterfall diagram of the measured shaft displacement with $5.9 \times 10^{-3}$ kg·m and $2\pi$ rad/s$^2$. N is the number of samples.

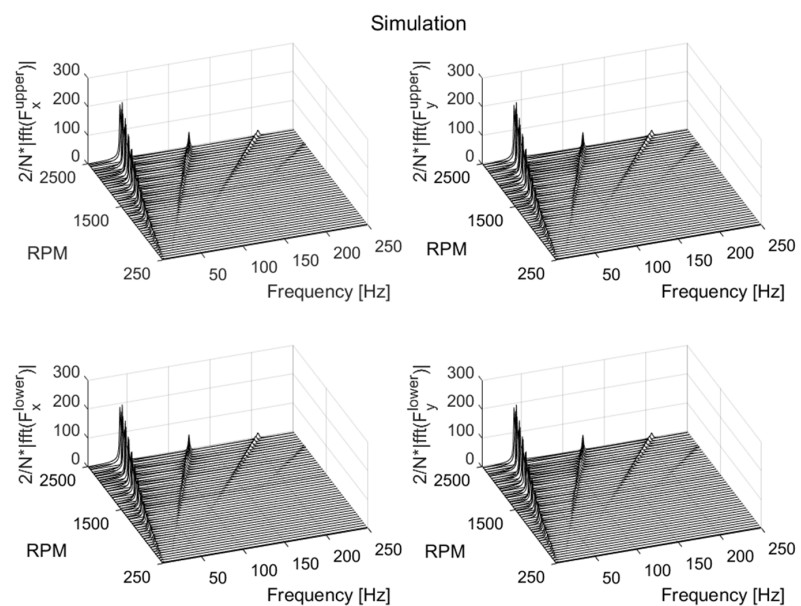

**Figure 25.** The waterfall diagram of the simulated bearing forces with $5.9 \times 10^{-3}$ kg·m and $2\pi$ rad/s$^2$. N is the number of samples.

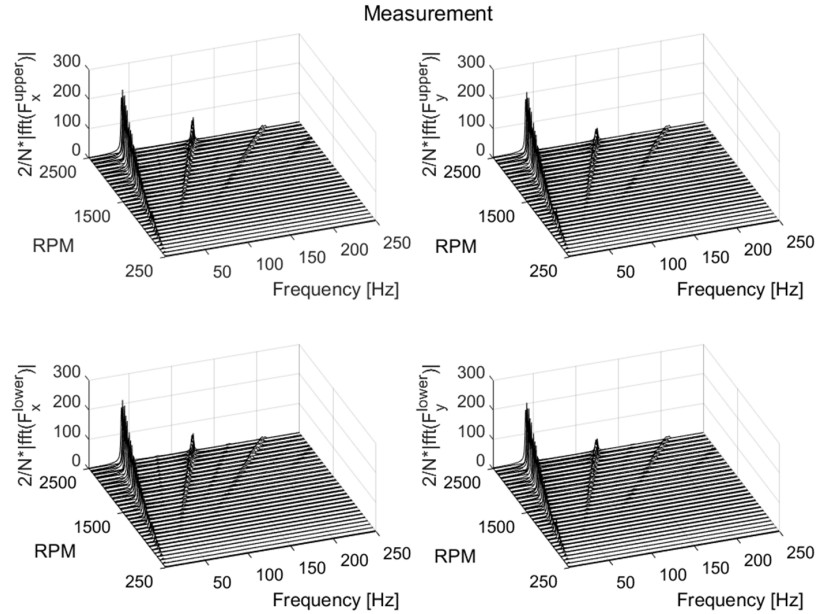

**Figure 26.** The waterfall diagram of the measured bearing forces with $5.9 \times 10^{-3}$ kg·m and $2\pi$ rad/s$^2$. N is the number of samples.

Furthermore, other frequency orders also appeared in the experiment, although their amplitudes were not as pronounced as in the $1 \times \Omega$ and $3 \times \Omega$ (and $5 \times \Omega$ in some cases) frequency orders. The frequency order at $7 \times \Omega$ is a multiple of the frequency due to the number of pads. All bearing pads were assumed to be similar, and there was no irregularity in the pad geometry. However, this is not true, and the design is approximated within a specified tolerance. Imperfections in the design can be a reason for the existence of other frequency orders. In the simulation, however, no other frequency other than $1 \times \Omega$, $3 \times \Omega$, $5 \times \Omega$, and $7 \times \Omega$ frequency orders existed for a frequency range between 3 Hz and 250 Hz.

## 5. Discussions and Conclusions

The dynamic responses of the vertical rotor with TPJBs under constant and variable rotor spin speeds were successfully modeled by predefining the bearing coefficients using polynomial equations and periodic functions (sine and cosine). These equations represent the direct and cross-coupling bearing coefficients as functions of the location of the journal center and rotor speed. For polynomial equations, the fitting was computed using MATLAB, and the *poly53* function was chosen based on the goodness-of-fit performance test. The response simulations under constant rotor speed were carried out for different unbalance magnitudes and rotor speeds, whereas the transient responses were simulated for different unbalance magnitudes and rotor angular accelerations. Both models were validated with experiments.

The orbits and bearing forces were square-shaped, and the simulation and experimental results displayed similar patterns. Both the magnitude and the shape of the responses were influenced by the unbalance magnitude and rotor speed. At lower unbalance magnitudes and rotor speeds, however, the bearings operated at low orbit amplitude, and the bearing dynamic coefficients were almost independent of the load direction. The square-shaped orbits and forces were not as amplified as those under higher unbalanced magnitude and rotor speed. Such as in the experiments, the FFT of the simulated responses contains amplitude peaks at the first $(1 \times \Omega)$, third $(3 \times \Omega)$, and fifth $(5 \times \Omega)$ frequency orders. The first frequency order $(1 \times \Omega)$ is due to the unbalance force, whereas the third and fifth frequency orders $(3 \times \Omega$ and $5 \times \Omega)$ are associated with the number of pads.

One can conclude that the simulation results compared favorably with the experimental results, with some minor deviations. Uncertainties in bearing parameter prediction were the main potential reason for these deviations. The fluid film lubrication simulation used assumptions, and bearing parameters were approximated. For instance, the geometry of the pads was assumed to be similar and uniform, which is, however, not true in the actual bearings. Furthermore, unlike in the simulations, the response measurements close to 1000 rpm were affected by rotor rig-induced vibration noise.

The simulation model presented in this paper is advantageous since it does not require solving the fluid film lubrication model at each time step. It was found to be more efficient than the classical simulation model in terms of simulation time and useful for heavy simulations that are impractical to solve with the standard numerical integration procedure.

**Author Contributions:** Conceptualization, G.B.B., R.G. and J.-O.A.; methodology, G.B.B., R.G. and J.-O.A.; software, G.B.B.; validation, G.B.B., R.G. and J.-O.A.; formal analysis, G.B.B., R.G. and J.-O.A.; investigation, G.B.B.; resources, G.B.B., R.G. and J.-O.A.; data curation, G.B.B., R.G. and J.-O.A.; writing—original draft preparation, G.B.B.; writing—review and editing, G.B.B., R.G. and J.-O.A.; visualization, G.B.B.; supervision, G.B.B., R.G. and J.-O.A.; project administration, G.B.B., R.G. and J.-O.A.; funding acquisition, R.G. and J.-O.A. All authors have read and agreed to the published version of the manuscript.

**Funding:** The research presented in this thesis was carried out as a part of "Swedish Hydropower Centre-SVC". SVC has been established by the Swedish Energy Agency, Energiforsk and Svenska Kraftnät together with Luleå University of Technology, KTH Royal Institute of Technology, Chalmers University of Technology, Uppsala University and Lund University. Participating companies and industry associations are the following: Andritz Hydro, Boliden, Fortum Sweden, Holmen Energi, Jämtkraft, Karlstads Energi, LKAB, Mälarenergi, Norconsult, Rainpower, Skellefteå Kraft, Sollefteåforsen, Statkraft Sverige, Sweco Sverige, Tekniska verken i Linköping, Uniper, Vattenfall R&D, Vattenfall Vattenkraft, Voith Hydro, WSP Sverige, Zinkgruvan, and AFRY.

**Data Availability Statement:** The data used to support the findings of this study are included within the article.

**Acknowledgments:** The authors would like to acknowledge the contribution the staff at Vattenfall R&D and David Rondon for their contribution during experimentation at Älvkarleby Vattenfall.

**Conflicts of Interest:** The authors declare no conflict of interest.

## Nomenclature

| | |
|---|---|
| $a_g$ | The length of the line connecting the center of the rotor and center of gravity (m) |
| $C_g$ | Center of gravity (-) |
| $C_{ij}^{max}$, $C_{ij}^{max}$ | Maximum, minimum bearing damping in the local ($i,j \to \xi,\eta$) coordinates (N-s/m) |
| $\mathbf{C_\beta}$ ($\mathbf{C_\beta^{upper}}$, $\mathbf{C_\beta^{lower}}$) | Bearing (upper, lower) damping matrix in the local $\xi$ and $\eta$ coordinates (N-s/m) |
| $\mathbf{C_B}$ | Bearing damping matrix in the Cartesian coordinates (N-s/m) |
| $d$ | The minimum distance between the center of the journal and the unbalance mass (m) |
| $e$ | Eccentricity (m) |
| $e_x$ | Eccentricity in the Cartesian x-direction (m) |
| $e_y$ | Eccentricity in the Cartesian y-direction (m) |
| $e_{exp}^i$, $e_{exp}^i$ | Measured, simulated mean amplitude of the orbit (for $i$: upper or lower bearing), (m) |
| $\mathbf{f_u}$ | Unbalance force vector (N) |
| $F_{exp}^i$, $F_{sim}^i$ | Measured, simulated mean force (for $i$: upper or lower bearing), (N) |
| $\mathbf{G}$ | Gyroscopic matrix |
| $K_{ij}^{max}$, $K_{ij}^{min}$ | Maximum, minimum bearing damping in the local ($i,j \to \xi,\eta$) coordinates (N/m) |
| $\mathbf{K_\beta}$ ($\mathbf{K_\beta^{upper}}$, $\mathbf{K_\beta^{lower}}$) | Bearing (upper, lower) stiffness matrix in the local $\xi$ and $\eta$ coordinates (N/m) |
| $\mathbf{K_B}$ | Bearing stiffness matrix in the Cartesian coordinates (N/m) |
| $\mathbf{K_S}$ | The stiffness of the bracket structure (N/m) |
| L | The sum of the square errors |
| $\mathbf{M}$ | Mass matrix (kg) |
| $m_u$ | Unbalance mass (kg) |
| $M_e$ | Number of elements in the axial direction (-) |
| $n$ | Number of pads (-) |
| $N_e$ | Number of elements in the circumferential direction (-) |
| R | Radius of the journal (m) |
| $R_{adj}^2$ | Adjusted R-square |
| $r$ | The polynomial degrees of the relative eccentricity (-) |
| $s$ | The polynomial degrees of the rotor speed (-) |
| $\mathbf{T}$ | Transformation matrix (-) |
| $T_{in}$ | The inlet temperature of the lubricant (supply lubricant) (°C) |
| $T_{out}$ | The outlet average temperature of the lubricant (°C) |
| $Y_k$ | Calculated maximum or minimum local bearing coefficient, values from RAPPID (N/m or N-s/m) |
| $\widetilde{Y}_k$ | An approximated maximum or minimum local bearing coefficient (N/m or N-s/m) |
| $\%E_{disp}^i$, $\%E_{Force}^i$ | Percentage errors of the amplitude of the orbit, (-) Percentage errors of the bearing force (-) |
| Greek symbols | |
| $\alpha$, $\dot{\alpha}$ | Eccentricity angle: the angle of the line connecting the center of the bearing and center of journal from the positive *x*-axis, whirling speed (rad, rad/s) |
| $\beta_{ij}$ | Regression coefficients |
| $\varepsilon$ | Relative eccentricity, $\varepsilon = e/C_b$ [-] |
| $\phi, \dot{\phi}, \ddot{\phi}$ | Angle of the line connecting the center of the journal and center of gravity of the journal from the positive *x*-axis, rotor spin speed, and rotor angular acceleration (rad, rad/s, rad/s²) |
| $\xi$ | Rotating coordinate that passes through the center of the bearing and collinear to the eccentricity line |
| $\eta$ | Rotating coordinate that is perpendicular to the eccentricity line |
| $\Omega$ | Constant angular speed of the journal (rad/s) |

## Appendix A

**Table A1.** Summary of the models.

| | | | Relative Eccentricity ($\varepsilon^i$) | | | | | | | | |
|---|---|---|---|---|---|---|---|---|---|---|---|
| | | | First Order ($\varepsilon^1$) | | Second Order ($\varepsilon^2$) | | Third Order ($\varepsilon^3$) | | Fourth Order ($\varepsilon^4$) | | Fifth Order ($\varepsilon^5$) | |
| | | | *RMSE%* [%] | $R^2_{adj}$ | *RMSE%* [%] | $R^2_{adj}$ | *RMSE%* [%] | $R^2_{adj}$ | *RMSE%* [%] | $R^2_{adj}$ | *RMSE%* [%] | $R^2_{adj}$ |
| **Rotor speed ($\Omega^i$)** | Second order ($\Omega^2$) | $K^{max}_{\xi\xi}$ | 60.8 | 0.543 | 43.3 | 0.768 | 23.7 | 0.93 | 11.3 | 0.984 | 4.7 | 0.997 |
| | | $K^{max}_{\xi\eta}$ | 58.2 | 0.587 | 40.1 | 0.803 | 20.6 | 0.948 | 9.2 | 0.989 | 3.5 | 0.998 |
| | | $K^{max}_{\eta\xi}$ | 57.5 | 0.592 | 39.6 | 0.806 | 20.2 | 0.949 | 9 | 0.989 | 3.5 | 0.998 |
| | | $K^{max}_{\eta\eta}$ | 29.2 | 0.838 | 17.3 | 0.943 | 6.2 | 0.992 | 2.2 | 0.999 | 1.0 | 0.999 |
| | | $C^{max}_{\xi\xi}$ | 50.5 | 0.588 | 25.4 | 0.895 | 11.4 | 0.978 | 4.4 | 0.996 | 1.4 | 0.999 |
| | | $C^{max}_{\xi\eta}$ | 50.6 | 0.631 | 23.7 | 0.919 | 9.9 | 0.985 | 3.6 | 0.998 | 1.2 | 0.999 |
| | | $C^{max}_{\eta\xi}$ | 50.9 | 0.63 | 23.8 | 0.918 | 10 | 0.985 | 3.6 | 0.998 | 1.2 | 0.999 |
| | | $C^{max}_{\eta\eta}$ | 20.7 | 0.802 | 5.9 | 0.983 | 1.8 | 0.998 | 0.6 | 0.999 | 0.5 | 0.999 |
| | | $K^{min}_{\xi\xi}$ | 28.9 | 0.839 | 16.9 | 0.944 | 5.8 | 0.993 | 2 | 0.999 | 1.6 | 0.999 |
| | | $K^{min}_{\xi\eta}$ | 56.9 | 0.599 | 39 | 0.811 | 19.8 | 0.951 | 8.8 | 0.99 | 3.3 | 0.998 |
| | | $K^{min}_{\eta\xi}$ | 57.7 | 0.593 | 39.6 | 0.807 | 20.2 | 0.949 | 9 | 0.989 | 3.4 | 0.998 |
| | | $K^{min}_{\eta\eta}$ | 0.7 | 0.999 | 0.6 | 0.999 | 0.6 | 0.999 | 0.6 | 0.999 | 0.6 | 0.999 |
| | | $C^{min}_{\xi\xi}$ | 21.0 | 0.797 | 6.2 | 0.981 | 2.2 | 0.997 | 1.7 | 0.998 | 1.7 | 0.998 |
| | | $C^{min}_{\xi\eta}$ | 50.6 | 0.633 | 23.5 | 0.92 | 9.9 | 0.985 | 3.6 | 0.998 | 1.3 | 0.999 |
| | | $C^{min}_{\eta\xi}$ | 50.4 | 0.633 | 23.5 | 0.92 | 9.8 | 0.985 | 3.6 | 0.998 | 1.4 | 0.999 |
| | | $C^{min}_{\eta\eta}$ | 0.8 | 0.93 | 0.8 | 0.929 | 0.8 | 0.926 | 0.8 | 0.921 | 0.8 | 0.915 |
| | Third order ($\Omega^3$) | $K^{max}_{\xi\xi}$ | 62.1 | 0.524 | 35.8 | 0.841 | 24 | 0.928 | 11.6 | 0.983 | 4.9 | 0.997 |
| | | $K^{max}_{\xi\eta}$ | 59.4 | 0.57 | 31.9 | 0.875 | 20.8 | 0.946 | 9.5 | 0.989 | 3.7 | 0.998 |
| | | $K^{max}_{\eta\xi}$ | 58.7 | 0.575 | 31.4 | 0.878 | 20.5 | 0.948 | 9.3 | 0.989 | 3.6 | 0.998 |
| | | $K^{max}_{\eta\eta}$ | 29.8 | 0.831 | 10.4 | 0.979 | 6.2 | 0.992 | 2.3 | 0.998 | 0.9 | 0.999 |
| | | $C^{max}_{\xi\xi}$ | 51.6 | 0.57 | 26.2 | 0.888 | 11.5 | 0.978 | 4.4 | 0.996 | 1.4 | 0.999 |
| | | $C^{max}_{\xi\eta}$ | 51.7 | 0.615 | 24.4 | 0.914 | 10 | 0.985 | 3.7 | 0.997 | 1.3 | 0.999 |
| | | $C^{max}_{\eta\xi}$ | 52 | 0.614 | 24.5 | 0.913 | 10.1 | 0.985 | 3.7 | 0.998 | 1.2 | 0.999 |
| | | $C^{max}_{\eta\eta}$ | 21.2 | 0.794 | 6.0 | 0.983 | 1.8 | 0.998 | 0.6 | 0.999 | 0.4 | 0.999 |
| | | $K^{min}_{\xi\xi}$ | 29.4 | 0.833 | 10.0 | 0.98 | 5.8 | 0.993 | 2.0 | 0.999 | 1.5 | 0.999 |
| | | $K^{min}_{\xi\xi}$ | 58.1 | 0.582 | 30.8 | 0.882 | 20.0 | 0.95 | 9.0 | 0.989 | 3.5 | 0.998 |
| | | $K^{min}_{\eta\xi}$ | 58.9 | 0.576 | 31.4 | 0.879 | 20.4 | 0.948 | 9.2 | 0.989 | 3.6 | 0.998 |
| | | $K^{min}_{\eta\eta}$ | 0.6 | 0.999 | 0.6 | 0.999 | 0.5 | 0.999 | 0.5 | 0.999 | 0.5 | 0.999 |
| | | $C^{min}_{\xi\xi}$ | 21.4 | 0.789 | 6.3 | 0.981 | 2.2 | 0.997 | 1.7 | 0.998 | 1.6 | 0.998 |
| | | $C^{min}_{\xi\eta}$ | 51.7 | 0.617 | 24.3 | 0.915 | 10.0 | 0.985 | 3.7 | 0.998 | 1.3 | 0.999 |
| | | $C^{min}_{\eta\xi}$ | 51.5 | 0.618 | 24.2 | 0.915 | 9.9 | 0.985 | 3.6 | 0.998 | 1.3 | 0.999 |
| | | $C^{min}_{\eta\eta}$ | 0.3 | 0.985 | 0.3 | 0.986 | 0.3 | 0.986 | 0.3 | 0.986 | 0.3 | 0.990 |

## Appendix B

**Table A2.** Calculated coefficients of the two-dimensional polynomial equation (*poly53*) as a function of centered and scaled relative eccentricity ($\varepsilon$) and rotor speed ($\Omega$). The relative eccentricity ($\varepsilon$) is normalized by mean and standard deviation of 0.4011 and 0.2589, respectively. The rotor speed ($\Omega$) is normalized by mean and standard deviation of 1292 and 803.5, respectively.

| | $K_{\xi\xi}^{max}$ | $K_{\xi\eta}^{max}$ | $K_{\eta\xi}^{max}$ | $K_{\eta\eta}^{max}$ | $C_{\xi\xi}^{max}$ | $C_{\xi\eta}^{max}$ | $C_{\eta\xi}^{max}$ | $C_{\eta\eta}^{max}$ |
|---|---|---|---|---|---|---|---|---|
| $\beta_{00}$ | 1.8E6 | 2.6E5 | 2.9E5 | 9.5E5 | 1E4 | 1.3E3 | 1.3E3 | 7.2E3 |
| $\beta_{10}$ | 2.3E6 | 5.1E5 | 5.1E5 | 6.1E5 | 9.7E3 | 2.3E3 | 2.3E3 | 3.1E3 |
| $\beta_{01}$ | 1.1E6 | 1.6E5 | 1.7E5 | 5.8E5 | −3.2E2 | −4.2E1 | −5.3E1 | −1.8E2 |
| $\beta_{20}$ | −8.5E5 | 3.9E4 | 5E4 | 3.1E5 | 2.3E3 | 9.3E2 | 9.3E2 | 1.6E3 |
| $\beta_{11}$ | −5E5 | 3E4 | 3.7E4 | 3.2E5 | −3.5E2 | −9.1E1 | −9.2E1 | −2.1E1 |
| $\beta_{02}$ | 3.2E3 | −3.1E2 | −1.3E3 | −1.8E4 | −1.2E2 | −0.4E0 | 2.6E0 | −1.2E2 |
| $\beta_{30}$ | −8.9E5 | −1.8E4 | −1E4 | 9.6E4 | 8.5E2 | 4.1E2 | 4.2E2 | 5.5E2 |
| $\beta_{21}$ | −6.9E5 | −4.6E3 | 1.6E3 | 1.6E5 | −3.1E2 | −9.4E1 | −8.4E1 | −1.8E1 |
| $\beta_{12}$ | 1.1E4 | −4.7E3 | −5.5E3 | −2.4E4 | 1.5E1 | −3.3E0 | −4.6E0 | 1.5E0 |
| $\beta_{03}$ | 1.4E3 | 1.2E3 | 1.4E3 | −9.9E3 | 8.6E1 | 1.3E1 | 1.1E1 | 2.1E1 |
| $\beta_{40}$ | 3E6 | 4.6E5 | 4.5E5 | 9.2E4 | 6.9E3 | 1.1E3 | 1.1E3 | 2.6E2 |
| $\beta_{31}$ | 2.7E6 | 4.6E5 | 4.5E5 | 1.3E5 | −3.2E2 | −5.7E1 | −5.8E1 | −7.2E1 |
| $\beta_{22}$ | −1.2E5 | −2.2E4 | −2.2E4 | −1.6E4 | −3E1 | −1.9E1 | −1.8E1 | −4.6E1 |
| $\beta_{13}$ | 1E4 | −1.8E2 | 5E2 | −6.5E3 | 1.5E2 | 2E1 | 2.2E1 | −3.2E1 |
| $\beta_{50}$ | 1.8E6 | 2.6E5 | 2.6E5 | 4E4 | 3.7E3 | 5.8E2 | 5.8E2 | 8.1E1 |
| $\beta_{41}$ | 1.9E6 | 2.9E5 | 2.8E5 | 6E4 | −1.4E2 | −1.2E1 | −1.5E1 | −4.4E1 |
| $\beta_{32}$ | −8.8E4 | −1.4E4 | −1.3E4 | −1.4E3 | −2.4E1 | −1.2E1 | −1.1E1 | −2.8E1 |
| $\beta_{23}$ | 6.9E3 | −7.9E2 | −8.5E2 | 2.1E3 | 9E1 | 5.1E0 | 4.4E0 | −3E0 |
| | $K_{\xi\xi}^{min}$ | $K_{\xi\eta}^{min}$ | $K_{\eta\xi}^{min}$ | $K_{\eta\eta}^{min}$ | $C_{\xi\xi}^{min}$ | $C_{\xi\eta}^{min}$ | $C_{\eta\xi}^{min}$ | $C_{\eta\eta}^{min}$ |
| $\beta_{00}$ | 9.5E5 | −2.7E5 | −2.5E5 | 5.6E5 | 7.2E3 | −1.2E3 | −1.3E3 | 5.1E3 |
| $\beta_{10}$ | 5.9E5 | −4.8E5 | −4.8E5 | 1.5E3 | 3.1E3 | −2.2E3 | −2.2E3 | −0.4E0 |
| $\beta_{01}$ | 5.9E5 | −1.6E5 | −1.5E5 | 3.4E5 | −9.6E1 | 6.4E1 | 5.4E1 | −1.8E2 |
| $\beta_{20}$ | 3.4E5 | −6.2E4 | −5E4 | 1.3E3 | 1.6E3 | −9.2E2 | −9.2E2 | −0.4E0 |
| $\beta_{11}$ | 3.1E5 | −4.5E4 | −3.7E4 | −4.1E2 | −2E2 | 9E1 | 8.9E1 | −2.2E1 |
| $\beta_{02}$ | −2.9E4 | −1E3 | −1.9E3 | −1.1E4 | −2.1E2 | −9.3E0 | −6.1E0 | −1.3E2 |
| $\beta_{30}$ | 1.6E5 | −2.3E3 | 6.5E3 | −1.5E3 | 6.9E2 | −4.3E2 | −4.4E2 | −2.7E0 |
| $\beta_{21}$ | 1.7E5 | −1.2E4 | −4.4E3 | −3.6E3 | −1.1E2 | 4.4E1 | 5E1 | −9.2E0 |
| $\beta_{12}$ | −4.9E4 | −7.9E2 | −8.9E2 | 1.7E2 | −1.8E2 | −3.2E1 | −3.5E1 | −2.4E0 |
| $\beta_{03}$ | −2.9E4 | −3.1E3 | −2.6E3 | −4.8E3 | −1E2 | −2E1 | −2.2E1 | 6.2E1 |
| $\beta_{40}$ | 7.5E4 | −4.1E5 | −4.2E5 | 3.8E2 | 2.4E2 | −1.1E3 | −1.1E3 | 1.2E0 |
| $\beta_{31}$ | 1.2E5 | −4.1E5 | −4.2E5 | 1.6E3 | −2.8E1 | 6.7E1 | 6.8E1 | −2E0 |
| $\beta_{22}$ | −1.8E4 | 2.2E4 | 2.2E4 | −1.3E3 | −2E1 | 1.6E1 | 1.7E1 | −4.2E0 |
| $\beta_{13}$ | −9.7E3 | −4E3 | −3.1E3 | −1E3 | −3.8E0 | −3.9E1 | −3.9E1 | 5.8E0 |
| $\beta_{50}$ | 1.2E4 | −2.3E5 | −2.4E5 | 1.5E3 | 1.5E1 | −5.5E2 | −5.5E2 | 0.5E0 |
| $\beta_{41}$ | 4.1E4 | −2.5E5 | −2.6E5 | −2.3E1 | −7.4E1 | 3.6E1 | 3.6E1 | 0.7E0 |
| $\beta_{32}$ | 6.1E3 | 1.6E4 | 1.6E4 | −8.5E2 | 5.5E1 | 2.8E1 | 2.8E1 | −0.6E0 |
| $\beta_{23}$ | 1.3E4 | −5.4E2 | −6.8E2 | 2.2E3 | 1E2 | −1.1E1 | −1.2E1 | 0.7E0 |

**Appendix C**

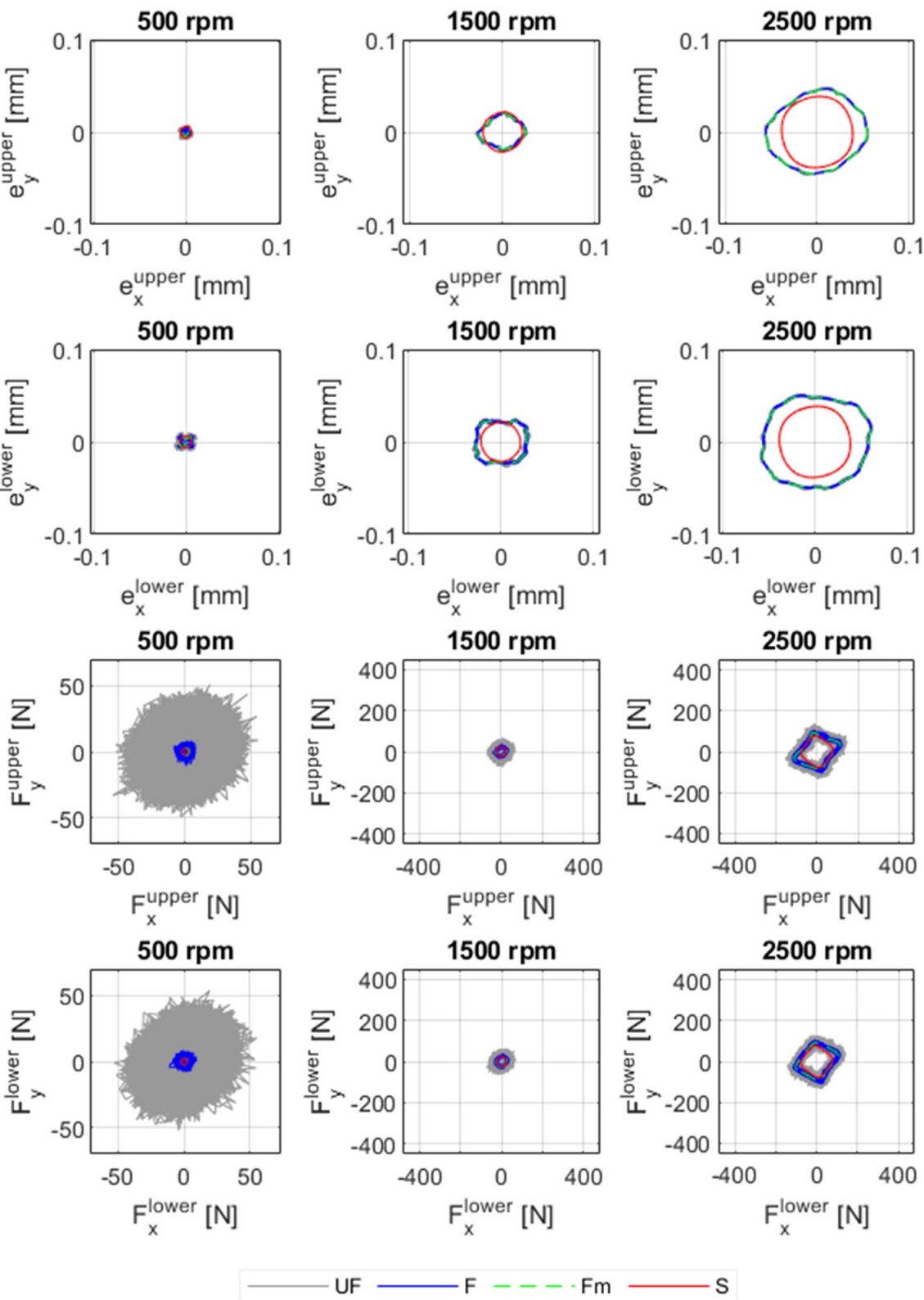

**Figure A1.** Orbits and bearing forces of the upper and lower bearings with $1.7 \times 10^{-3}$ kg·m unbalance magnitude and at 500 rpm, 1500 rpm, and 2500 rpm. UF: unfiltered measurement signal; F: filtered measurement signal by a band pass filter; S: simulation.

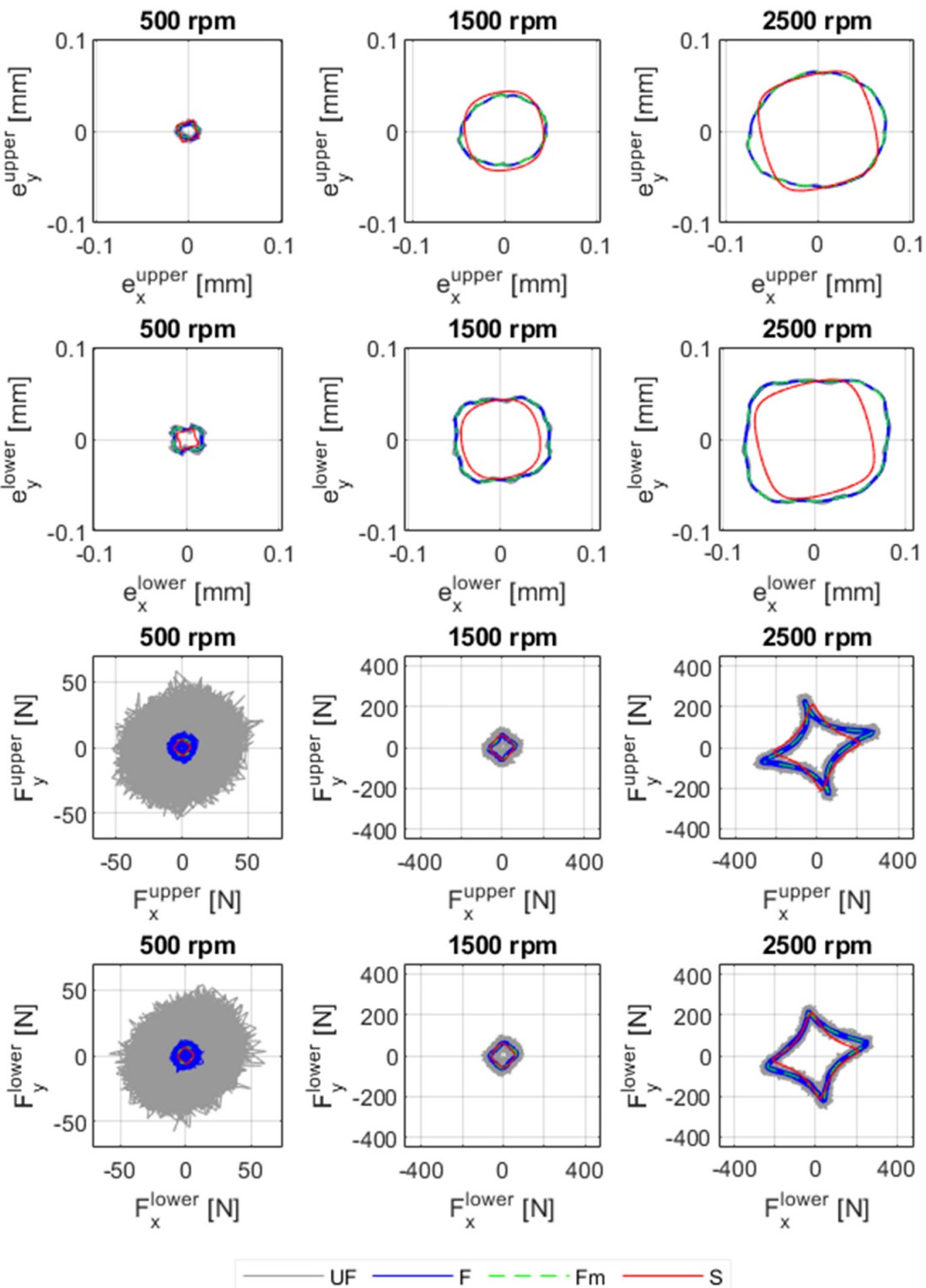

**Figure A2.** Orbits and bearing forces of the upper and lower bearings with $3.8 \times 10^{-3}$ kg·m unbalance magnitude and at 500 rpm, 1500 rpm, and 2500 rpm. UF: unfiltered measurement signal; F: filtered measurement signal by a band pass filter; S: simulation.

### Appendix D. Fluid Film Forces and FEM

A plain cylindrical journal bearing without grooves is considered. The schematic representation of the plain cylindrical journal bearing and the bearing parameters are shown in Figure A3 and Table A3, respectively. The fluid film pressure distribution within the gap between the shaft and bearing is modeled using the classical Reynolds equation as shown in Equation (A1). The lubricant is assumed to be an iso-viscous and incompressible fluid with constant density. The flow is assumed to be laminar.

$$\frac{1}{R^2}\frac{\partial}{\partial\psi}\left(h^3\frac{\partial p}{\partial\psi}\right) + \frac{\partial}{\partial z}\left(h^3\frac{\partial p}{\partial z}\right) = 6\mu\Omega\frac{\partial h}{\partial\psi} + 12\mu\frac{\partial h}{\partial t} \tag{A1}$$

where $p$ is the fluid-film pressure, $\mu$ is the viscosity of the lubricant, $\Omega$ is the angular speed of the rotor, $\psi$ is the circumferential coordinate and $z$ is the axial coordinate. The fluid film thickness, $h(\psi)$, is given as a function of circumferential coordinate ($\psi$) as shown in Equation (A2).

$$h(\psi) = C_p + e\,\cos(\psi - \alpha) = C_p + e_x\,\cos(\psi) + e_y\,\sin(\psi) \tag{A2}$$

where $C_p$ is the radial bearing clearance, e is eccentricity, $e_x$ and $e_x$ represent journal eccentricities in the stationary $x$ and $y$ axes. The bearing forces are calculated by integrating the pressure distribution over the fluid film domain, Equation (A3).

$$\begin{bmatrix} F_x \\ F_y \end{bmatrix} = \int_0^{2\pi}\int_0^L p\begin{bmatrix} \cos\psi \\ \sin\psi \end{bmatrix}(Rd\psi)dz \tag{A3}$$

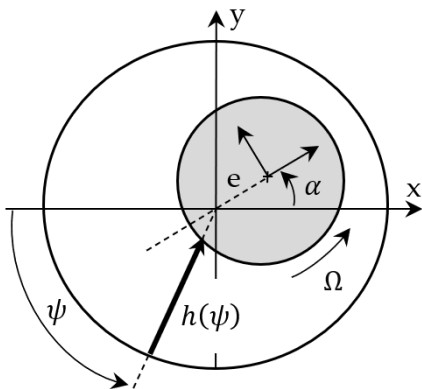

**Figure A3.** Schematic representation of a plain cylindrical journal bearing.

**Table A3.** Plain cylindrical journal bearing parameters.

| Descriptions | Values |
| --- | --- |
| Diameter (mm) | 49.815 |
| Length (mm) | 20 |
| Radial bearing clearance (μm) | 130 |
| Rotor speed (rpm) | 1500 |
| Lubricant | Q6 Handel oil |
| Oil supply pressure (MPa) | 0.01 |
| Viscosity (mPa·s) | 46.11 |

The Reynolds equation governing the fluid flow is numerically solved. A finite element analysis is used to calculate the pressure distribution within the fluid film. The fluid film domain is discretized by $M_e \times N_e$ mesh elements, and each element is represented by a four-node bilinear quadrilateral shape function. The FEM equation in matrix form is shown in Equation (A4) with boundary conditions of zero pressure at the sides of the bearing.

$$\left[\mathbf{K_P}\right]\{\mathbf{p}\} = [\mathbf{K_{U_x}}]\{\mathbf{U_x}\} + [\mathbf{K_{\dot{h}}}]\{\dot{\mathbf{h}}\} \tag{A4}$$

where $\left[\mathbf{K_P}\right]$ is the pressure fluidity matrix, $[\mathbf{K_{U_x}}]$ is the shear fluidity matrix, and $[\mathbf{K_{\dot{h}}}]$ is the squeeze component matrix. The bearing is symmetrical along the axial direction, and therefore, only half of the bearing, which is 10 mm in the axial direction, is considered in the analysis to reduce the computation time.

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
