# Peer review of "Speed-Dependent Bearing Models for Dynamic Simulations of Vertical Rotors"

_machines, doi:10.3390/machines10070556_

Round 1
Reviewer 1 Report
The manuscript on the dynamic simulations on tilting pad and vertical rotors could be interesting.
However, I recommend the authors to read some papers of significance on this topic and published during the last 40 years.
Both linear and nonlinear behaviors of fixed geometry and tilting pad journal bearings subjected to unbalance loads have been greatly presented and discussed. More precisely, more advanced modellings were developed like EHD numerical code (including mechanical pad deformations and/or effective (operating) temperature) by Desbordes et al (ASME Journal of Tribology) in the nineties and more recently, about 25 years ago, TEHD theoretical models (considering both thermal and mechanical deformations as well as local temperature (i.e. viscosity) variations) by Monmousseau et al (ASME Journal of Tribology & ASME Journal of Engineering for Gas Turbines and Power).
It is also noted that a lot of information is missing.
Main recommendations are the followings:
1) Read some papers on journal bearings in order to be more familiar with the technical and appropriate wordings.
2) Complete the review introduction with other papers of interest on the topic treated by the authors. Then, it is necessary to specify the originality this study and the limitations of such bearing model. Do not forget to note that, the calculations are fast but also, any new bearing calculations need preliminary calculations of the full range of eccentricity ratios and of rotational speeds.
3) In their introduction, the authors wrote “In contrast, vertical rotors have no stationary bearing operation point, and the rotor spins around the bearing continuously.”. This affirmation is generally wrong, even if sometimes it could be correct (for example, for a plain cylindrical journal bearing without feeding groove). For example, the 4 shoes tilting pad journal bearing studied by the authors has a static equilibrium position, even if no external load is applied: this is due to the geometrical preload of the bearing as well as due to the offset ratio of the pad pivots: the centered journal location in the bearing is a stationary journal position. The main difference between the horizontal rotor and the vertical rotor is that the weight of the rotor is supported by the journal bearings for the first configuration (radial loading) while the weight of the rotor is supported by the thrust bearing for the second configuration (axial loading).
4) The main assumptions should be given: isothermal regime (thermal effects are ignored), all bearing components are rigid (i.e. thermal and mechanical deformations are also ignored), linear dynamic coefficients (stiffness and damping coefficients) are used …..
5) The writing is too approximative: either you are using Reynolds equation or Navier-Stokes equations. Please, clarify this issue in the text as well as on figure(s) (like on figure 4, presenting the flow charts). If you are using or Navier-Stokes equations, justify why you are using these equations.
6) The boundary conditions on pressure at the limits of the pads should be given. What are the conditions on film rupture (cavitation zone), if there is any such a domain?
7) For the feeding pressure, it is preferable to use the SI unit “Pa” or “MPa” instead of “bar”.
8) A very important point is that the results presented and discussed by the authors are strongly dependent on the dynamic viscosity. However, the authors have never specified a value of viscosity which is not acceptable in a scientific paper of high quality. The name of the “oil” is not sufficient (by the way, it was impossible to find a corresponding viscosity of “Q6 Handel oil” on the Web site). Without this information, it is impossible to reproduce the numerical simulations, and so, no verification is possible.
9) In addition, the viscosity of the oil is greatly varying with temperature and so, the viscosity-temperature law should be given or, at least, the viscosity values should be given at 2 different temperatures (usually, at 40 and 100°C).
10) What is the temperature(s) or/and the viscosity value(s) used by the authors for their numerical simulations?
11) Is the temperature (viscosity) assumed to be constant for the full range of speeds, for both Models I & II? Please, note ere maybe some limitations due to the fact that oil film temperature increases with the increase of rotational speed.
12) I think there is a misunderstanding concerning the variable “alpha” (load angle). The simulations with model I are performed for a given relative eccentricity: the authors should explain how the direction of load could be imposed during these calculations? Isn’t it the angle of the journal center location in the bearing (and so, in that case “alpha” should not be named “load angle”). Clear explanations should be given!
13) Please, note that “eccentricity” expressed in percentage (%) is unusual in scientific or even technical papers. Either the eccentricity is express in “µm” or “mm” for the “real” (dimensional) eccentricity or without any unit for the “relative” (non-dimensional) eccentricity. You are using a non-dimensional eccentricity but you did not specify if this is relatively to the radial pad clearance (manufactured radial clearance) or to the radial bearing clearance (assembled radial clearance).
14) Concerning the experimental device and the experiments, the authors wrote “The bearings were supplied with 0.1 bar lubricant (Q6 Handel oil), which its temperature ranged from 23 °C to 33 °C in the test.” Please, specify the type of oil feeding: fully flooded pads, directed lubrication or leading-edge groove (LEG)?
15) In the same sentence, you should specify which temperature is ranged from 23 °C to 33 °C in the test” (test should be written with “s” => tests). Is it the feeding temperature? Oil discharge temperature? Average temperature? Pad or babbitt temperature? Please, also indicate where is measured the temperature in the bearing? Are you using thermocouple(s)? What is the accuracy of the measurements?
16) On figure 3, the global and the local (for solving Reynolds equation) coordinate systems should be drawn.
17) In Table 2 (maybe also in the text), “Journal (rotor) diameter (mm)” should be “Journal diameter (mm)”, “Axial pad length (mm)” should be “Pad length (mm)” or “Pad width (mm)”, “Circumferential pad length (degree)” should be “Angular pad amplitude (degree)” or “Pad angle (degree)”, “Radial distance between pads (degree)” should be written “Angular pivot position (degree)” (and write the 4 locations. For example, 0, 90, 180 and 270°).
18) In Table 2, the value of the “Radial bearing clearance” of 130 µm is very large and unusual for this side journal bearing. Explain why the authors have chosen such a large radial bearing clearance for performing their tests?
19) In Table 2, “Radial pad clearance” and “geometrical preload ratio” are missing.
20) In the models, there is no information concerning the pad equilibrium as well as no information on the tilt angle of the pads?
21) How is determined the attitude angle? Figures of evolution of “Load carrying capacity” and of “Attitude angle” should be presented versus angle “alpha”.
22) It seems that the numbers of equations in captions of Figure 5 (… according to Eqs. 3–6.) and of Figure 6 (… according to Eqs. 7–10.) are not correct.
23) There is a problem on wording: “Displacement orbit” should be either “orbit” or “trajectory of the journal center”.
24) The thickness of the pad or/and the mass of the pad or/and the material density of the pad should be given in a Table.
25) Explain how the mass of the pads is considered in the dynamic behavior of the bearings (calculations of the stiffness and damping coefficients) and in the dynamic behavior of the rotor?
26) The total mass of the rotor should be given.
27) The distance between the two journal bearings should be also given.
28) There is no information about the accuracy of measurements and operating variables: on the journal displacements, on the rotational speed, on the applied load, on temperature, …
29) Note that “Kgm” should be “kg.m” or “kg m”, knowing that “K” means Kelvin.
30) “1 cycle/s2 angular acceleration”: “cycle” is not an angle, so it could be better to use the SI unit “radian” => rd/s2.
31) Please, note that “LOP” and “LBP” are the abbreviations used when an external (radial and static) load is applied on pad (meaning on the pivot location) and between pad (usually symmetrically between two pivots), respectively. The use of these abbreviations “LOP” and “LBP” by the authors in this manuscript are often or always not correct. For example, the sentence “At lower eccentricities, the different between the LOP and LBP bearing coefficients are insignificant.” is not correct. In fact, in the context, “at lower eccentricities” means “at low orbit amplitudes”, “the different” should be “the difference”, “the LOP and LBP bearing coefficients” should be “the bearing dynamic coefficients are almost independent on the load direction” (or something similar). In addition, as there is no static loading, the load direction is always changing from 0 to 360 deg. like the angular location of the masse “m” (unbalance mass).
32) At the beginning, it should be clearly specified that there is no static radial load applied to the bearings, only a dynamic loading (synchronous unbalance load or pure rotating load) is considered in this study.
33) A nomenclature with a clear definition and units of all variables and parameters used in this manuscript is needed.
34) The authors wrote this first sentence in the last section “5. Discussions and Conclusions”: “The steady-state and non-stationary responses of the vertical rotor with TPJBs were successfully modelled by predefining the bearing coefficients using polynomial equations.” No result and no discussion on steady-state regime have been presented in this manuscript, only some dynamic coefficients have been calculated for given (imposed) eccentricities.
35) The authors wrote this sentence in the last section “5. Discussions and Conclusions”: “These equations were derived from the two-dimensional polynomial regression, and the bearing coefficients were represented as functions of eccentricity ratio and rotor speed.” Note that the dynamic coefficients of the bearing are also dependent of the angle “alpha”, or of the angular location of the journal center, not only of journal eccentricity.
36) The authors wrote this sentence in the last section “5. Discussions and Conclusions”: “Steady-state response simulations were carried out for different unbalance magnitudes and rotor speeds, …” The steady-state regime is obtained for a constant radial load, constant in both magnitude and direction leading to a unique journal center position. When an unbalance loading is applied, the journal center location is varying: it describes a closed orbit or a trajectory and so, it could be a cycling regime (dynamic regime) or a transient regime (when the rotational speed is varying, for example) but it is not at all a steady-state regime (permanent regime).
37) The authors wrote this sentence in the last section “5. Discussions and Conclusions”: “Since the bearing properties were different at LOP and LBP, the four pad bearing generated square-shaped orbits and forces at large unbalance magnitudes.” However, for the above reason (see comment 31), “different at LOP and LBP” is not correct and not appropriate in this sentence; please, correct it. It is also better to write “the four-shoes tilting pad journal bearing” instead of “the four pad bearing”. This result “generated square-shaped orbits” is not a new result as it was previously obtained by numerous other researchers: a triangle-shape orbit by Desbordes for the three-shoes tilting-pad journal bearing and a square-shape orbit by Monmousseau for the four-shoes tilting-pad journal bearing, for examples.
38) The authors wrote this sentence in the last section “5. Discussions and Conclusions”: “… the difference between the bearing coefficients at LOP and LBP were insignificant.”. This is not correct, see my previous comment (31).
39) The authors wrote this sentence in the last section “5. Discussions and Conclusions”: “The square-shaped displacement orbits …”. This is not correct, see my previous comment (23).
40) General comment: all the text, the captions and the tables should be checked in order to correct the manuscript by considering all the above comments. I cannot spend all my time by checking and correcting all mistakes/errors/ incorrectness.
In conclusion, a serious, major and strong revision of the manuscript is requested following the above mandatory comments/recommendations.
Author Response
Response to Reviewer 1 Comments
Dear Reviewer,
First of all, we would like to thank you for your very interesting comments and feedback. The authors tried to answer the comments as clear and as comprehensive as possible. Based on the reviewer’s feedback, some more explanations are included in the manuscript. Please let us know if there is anything unclear or needs more clarifications.
Further changes on the manuscript:
New viscosity value of the lubricant (i.e. 46.11 mPa⋅s) has been used in FEM of Model II. The value was taken from a commercial software (RAPPID) for Model I, which uses Vogel’s law and calculate the oil viscosity for a given temperature. As a result, Figure 15, Figure 16, and Table 5 may look slightly different from those in the first draft of the manuscript. No significant change has been observed on the computational time (Figure 15 & Table 5) and bearing force (Figure 16 & Table 5), although the trajectory of the journal center increased by around 12%. However, the change does not affect the conclusion or discussion of the manuscript as the main objective is to compare the computational efficiency of the two models.
The followings are notes and explanations for the review comments and feedbacks.
The manuscript on the dynamic simulations on tilting pad and vertical rotors could be interesting.
However, I recommend the authors to read some papers of significance on this topic and published during the last 40 years.
Both linear and nonlinear behaviors of fixed geometry and tilting pad journal bearings subjected to unbalance loads have been greatly presented and discussed. More precisely, more advanced modellings were developed like EHD numerical code (including mechanical pad deformations and/or effective (operating) temperature) by Desbordes et al (ASME Journal of Tribology) in the nineties and more recently, about 25 years ago, TEHD theoretical models (considering both thermal and mechanical deformations as well as local temperature (i.e. viscosity) variations) by Monmousseau et al (ASME Journal of Tribology & ASME Journal of Engineering for Gas Turbines and Power).
It is also noted that a lot of information is missing.
Main recommendations are the followings:
Point 1: Read some papers on journal bearings in order to be more familiar with the technical and appropriate wordings.
Response 1: Noted.
Point 2: Complete the review introduction with other papers of interest on the topic treated by the authors. Then, it is necessary to specify the originality this study and the limitations of such bearing model. Do not forget to note that, the calculations are fast but also, any new bearing calculations need preliminary calculations of the full range of eccentricity ratios and of rotational speeds.
Response 2: Noted and the following sentences are added to the manuscript.
Line 39: “For over the past 50 years, TPJBs were investigated both theoretically and experimentally, and research from Nicholas et al [2], Jones et al [3] and Keith [4] were among the earlier work to study the effect of various TPJB parameters on the dynamic bearing coefficients. Other researchers, like Lund et al [5, 6] and Someya [7], made remarkable contribution on journal bearing design, and presented the eight stiffness and damping coefficients of different types of journal bearings (including TPJBs), which are still being used today as a design guideline for many journal bearings. Besides, their scientific work and findings contributed for the advancement of modern theoretical models. Recent theoretical studied dealing with the dynamics of both fixed geometry and TPJBs use advance modelling, by taking into account many different factors. These include the mechanical deformation [8–10], thermal effect [8,11,12], pad flexibility [13–15] and pivots flexibility [16–18].”
Line 67: “Many attempts have been made by researchers to simplify and improve the computation efficiency of the simulation procedure. Perez et al. [28] used a simplified analytical bearing force expression to model the dynamics of a vertical rotor with hydrodynamic journal bearing. This approach has previously been presented by Yu Huang et al. [29] to estimate the instability threshold speed of cylindrical journal bearings.”
Line 79: “The present work proposes a simplified and computationally efficient model for calculating dynamic response simulation of a rotor with journal bearings. …. One limitation with the proposed model is the preliminary computational effort required to develop the bearing models.”
Point 3: In their introduction, the authors wrote “In contrast, vertical rotors have no stationary bearing operation point, and the rotor spins around the bearing continuously.”. This affirmation is generally wrong, even if sometimes it could be correct (for example, for a plain cylindrical journal bearing without feeding groove). For example, the 4 shoes tilting pad journal bearing studied by the authors has a static equilibrium position, even if no external load is applied: this is due to the geometrical preload of the bearing as well as due to the offset ratio of the pad pivots: the centered journal location in the bearing is a stationary journal position. The main difference between the horizontal rotor and the vertical rotor is that the weight of the rotor is supported by the journal bearings for the first configuration (radial loading) while the weight of the rotor is supported by the thrust bearing for the second configuration (axial loading).
Response 3: Noted and corrected accordingly.
Yes. We agree. The statements are misleading. The following sentences are removed from the manuscript.
“Horizontally orientated rotors usually have a stationary bearing operation point due to the dead weight of the rotor, and the dynamic coefficients of the bearing have to be calculated at this point by numerically solving the fluid film lubrication model. In contrast, vertical rotors have no stationary bearing operation point, and the rotor spins around the bearing continuously due to the unbalance mass.”
Point 4: The main assumptions should be given: isothermal regime (thermal effects are ignored), all bearing components are rigid (i.e. thermal and mechanical deformations are also ignored), linear dynamic coefficients (stiffness and damping coefficients) are used …..
Response 4: Noted and updated accordingly. Main assumptions used in the simulation model are mentioned in the manuscript
Line 174 “The linearized stiffness and damping coefficients ….”
Line 189 “All bearing components were assumed rigid, and the thermal and mechanical deformations were ignored. No friction between the surface of the pads and the bearing housing was considered..”
Point 5: The writing is too approximative: either you are using Reynolds equation or Navier-Stokes equations. Please, clarify this issue in the text as well as on figure(s) (like on figure 4, presenting the flow charts). If you are using or Navier-Stokes equations, justify why you are using these equations.
Response 5: Noted and updated accordingly.
- Model I: the bearing coefficients were calculated using a commercial software (RAPPID), which compute the fluid-flow governed by Navier-Stokes equations.
- Model II: the fluid-film lubrication model was described by Reynolds equation and solved by using MATLAB. In this paper, Model II was presented only to evaluate the computational efficiency of Model I, and the accuracy of this model was not as important as the computational efficiency. Therefore, Reynolds equation is used to simply the simulation.
The following sentences are included in the manuscript.
Line 163 “… (Model II) … . For simplification, the pressure distribution of the fluid film lubrication model was calculated by solving Reynolds equation. The simulation results from Model II are presented in Section 4.1.3. The computational efficiency of Model II was compared with that of the proposed model (Model I) based on the computational time.”
Point 6: The boundary conditions on pressure at the limits of the pads should be given. What are the conditions on film rupture (cavitation zone), if there is any such a domain?
Response 6: Noted and updated accordingly. The following sentences are included in the manuscript.
Line 194 “… Axial grooves between two consecutive pads were continuously fed by the supply lubricant and mixed with hot oil carried over from the preceding pad. For all simulations, the pressure and temperature of the supplied lubricant were 0.01 MPa and 23 °C, respectively. Table 3 shows the maximum temperature of the lubricant at each pad for different rotor speeds and relative eccentricity equal to 0.6. The viscosity of the lubricant was calculated by the program, which uses Vogel's Law to curve fit the data. The pressure and temperature of the lubricant at the side edges of the pads were assumed to be equal to 0.001 MPa and 23 °C. Besides, cavitation was assumed to occur when the fluid film pressure was less or equal to 0 Pa.”
Point 7: For the feeding pressure, it is preferable to use the SI unit “Pa” or “MPa” instead of “bar”.
Response 7: Noted and updated accordingly.
Point 8: A very important point is that the results presented and discussed by the authors are strongly dependent on the dynamic viscosity. However, the authors have never specified a value of viscosity which is not acceptable in a scientific paper of high quality. The name of the “oil” is not sufficient (by the way, it was impossible to find a corresponding viscosity of “Q6 Handel oil” on the Web site). Without this information, it is impossible to reproduce the numerical simulations, and so, no verification is possible.
Response 8: Noted and updated accordingly.
Line 196: “For all simulations, the pressure and temperature of the supplied lubricant was 0.01 MPa and 23 °C, respectively. Table 3 shows the maximum temperature of the lubricant at each pad for different rotor speed and relative eccentricity equal to 0.6. Viscosity of the lubricant was calculated by the program, which uses Vogel's Law to curve fit the data. The pressure and temperature of the lubricant at the side edges of the pads were assumed to be equal to 0.001 MPa and 23 °C.”
Point 9: In addition, the viscosity of the oil is greatly varying with temperature and so, the viscosity-temperature law should be given or, at least, the viscosity values should be given at 2 different temperatures (usually, at 40 and 100°C).
Response 9: Noted and updated accordingly. See Table 2.
Point 10 What is the temperature(s) or/and the viscosity value(s) used by the authors for their numerical simulations?
Response 10: Noted and updated accordingly. See comment 8
Point 11: Is the temperature (viscosity) assumed to be constant for the full range of speeds, for both Models I & II? Please, note ere maybe some limitations due to the fact that oil film temperature increases with the increase of rotational speed.
Response 11: The temperature of the lubricant varied with the rotor speed for Model I. See (10). See Table 3. For Model II, however, only one single temperature was considered since simulations were carried out just only for one rotor speed (i.e. 1500 rpm).
Point 12: I think there is a misunderstanding concerning the variable “alpha” (load angle). The simulations with model I are performed for a given relative eccentricity: the authors should explain how the direction of load could be imposed during these calculations? Isn’t it the angle of the journal center location in the bearing (and so, in that case “alpha” should not be named “load angle”). Clear explanations should be given!
Response 12: It is true that “alpha” should not be named as “load angle”. It is named an “eccentricity angle” instead.
Point 13: Please, note that “eccentricity” expressed in percentage (%) is unusual in scientific or even technical papers. Either the eccentricity is express in “µm” or “mm” for the “real” (dimensional) eccentricity or without any unit for the “relative” (non-dimensional) eccentricity. You are using a non-dimensional eccentricity but you did not specify if this is relatively to the radial pad clearance (manufactured radial clearance) or to the radial bearing clearance (assembled radial clearance).
Response 13: Noted and updated accordingly.
Point 14: Concerning the experimental device and the experiments, the authors wrote “The bearings were supplied with 0.1 bar lubricant (Q6 Handel oil), which its temperature ranged from 23 °C to 33 °C in the test.” Please, specify the type of oil feeding: fully flooded pads, directed lubrication or leading-edge groove (LEG)?
Response 14: Noted and the following sentence added to the manuscript.
Line 132 “The bearings were supplied with 0.01 MPa lubricant (Q6 Handel oil) and operated under fully flooded lubrication condition.”
Point 15: In the same sentence, you should specify which temperature is ranged from 23 °C to 33 °C in the test” (test should be written with “s” => tests). Is it the feeding temperature? Oil discharge temperature? Average temperature? Pad or babbitt temperature? Please, also indicate where is measured the temperature in the bearing? Are you using thermocouple(s)? What is the accuracy of the measurements?
Response 15: Noted and corrected accordingly.
Line 133: During each test, the inlet and outlet lubrication temperature were measured by PT100 thermocouples (with +0.5 °C accuracy). The temperature sensors were installed right before and after the upper and lower TPJBs. As shown in Figure 4, the inlet and outlet average temperature measurements are plotted as a function of rotor speed. .
Point 16: On figure 3, the global and the local (for solving Reynolds equation) coordinate systems should be drawn.
Response 16: Noted and update accordingly. Figure 3 is plotted with the global (X & Y) and local coordinates (ξ & η).
Point 17: In Table 2 (maybe also in the text), “Journal (rotor) diameter (mm)” should be “Journal diameter (mm)”, “Axial pad length (mm)” should be “Pad length (mm)” or “Pad width (mm)”, “Circumferential pad length (degree)” should be “Angular pad amplitude (degree)” or “Pad angle (degree)”, “Radial distance between pads (degree)” should be written “Angular pivot position (degree)” (and write the 4 locations. For example, 0, 90, 180 and 270°).
Response 17: Noted and updated accordingly.
Point 18: In Table 2, the value of the “Radial bearing clearance” of 130 µm is very large and unusual for this side journal bearing. Explain why the authors have chosen such a large radial bearing clearance for performing their tests?
Response18: Noted and updated accordingly
Line 130 “The parameters were chosen so that the Sommerfeld number of the bearings resembles those in hydropower generators.”
Point 19: In Table 2, “Radial pad clearance” and “geometrical preload ratio” are missing.
Response 19: Noted and updated accordingly. See Table 2
Point 20: In the models, there is no information concerning the pad equilibrium as well as no information on the tilt angle of the pads?
Response 20: Noted and updated accordingly. For this comment, we are not exactly sure if the following statement answers the reviewer’s question. But please let us know if we need to clarify more.
Line 192 “The pad tilts from its neutral positions by rolling. This means, the pivot contact point translates from its neutral position without sliding. In the neutral position, the four pivots were located at 0°, 90°, 180° and 270°. ”
Point 21: How is determined the attitude angle? Figures of evolution of “Load carrying capacity” and of “Attitude angle” should be presented versus angle “alpha”.
Response 21: The bearing coefficients were calculated using a commercial software RAPPID by predefining the location of the journal (relative eccentricity and eccentricity angle). No static equilibrium position of the journal is determined through iterative scheme, and therefore no attitude angle is calculated. However, the program calculates the bearing reaction load vectors (bearing reaction load and reaction load angle) at one predefined location of the journal center.
The following figure shows the bearing reaction load as a function of α for ξ = 0.5 and Ω = 1500 rpm. The reaction load angle and the eccentricity angle (α) are similar.
Point 22: It seems that the numbers of equations in captions of Figure 5 (… according to Eqs. 3–6.) and of Figure 6 (… according to Eqs. 7–10.) are not correct.
Response 22: Noted and updated accordingly.
Point 23) There is a problem on wording: “Displacement orbit” should be either “orbit” or “trajectory of the journal center”.
Response 23: Noted and updated accordingly.
Point 24: The thickness of the pad or/and the mass of the pad or/and the material density of the pad should be given in a Table.
Response 24: Noted and updated accordingly. See Table 2.
Point 25: Explain how the mass of the pads is considered in the dynamic behavior of the bearings (calculations of the stiffness and damping coefficients) and in the dynamic behavior of the rotor?
Response 25: Noted and updated accordingly.
Line 191 “Besides, the mass properties of the pad are considered in the dynamic solution that determines the transfer function.”
Point 26: The total mass of the rotor should be given.
Response 26: Noted and updated accordingly. See Table 1.
Point 27: The distance between the two journal bearings should be also given.
Response 27: Noted and updated accordingly. See Figure 2.
Point 28: There is no information about the accuracy of measurements and operating variables: on the journal displacements, on the rotational speed, on the applied load, on temperature, …
Response 28: Note and updated accordingly. The following sentences are added to the manuscript
Line 114: The displacement of the rotor at each bearing was measured by four 4 mm inductive proximity displacement sensors (Contrinex DW-AD-509-M8) mounted on the bearing housing. The sensors were calibrated in site for specific target material, which is steel in this case. Furthermore, a full Wheatstone bridge strain gauge (Kyowa KFG-5-350-D16-11L3M2S) was mounted on each bracket to measure the bearing reaction forces. Similarly, the sensors were calibrated in site both directly (shunt calibration) and indirectly (by applying a known force). Angular speed of the rotor was measured by an optical sensor with about + 1 rpm accuracy. All sensors were calibrated on situ by certified calibration equipment. A universal MX840 amplifier of HBM Quantum data acquisition system, and catman data acquisition software were used for measurement. The technical specification of the rotor rig is given in Table 1, and a further description is available in [35].
Point 29: Note that “Kgm” should be “kg.m” or “kg m”, knowing that “K” means Kelvin.
Response 29: Noted and updated accordingly.
Point 30: 1 cycle/s2 angular acceleration”: “cycle” is not an angle, so it could be better to use the SI unit “radian” => rd/s2.
Response 30: Noted and updated accordingly.
Point 31: Please, note that “LOP” and “LBP” are the abbreviations used when an external (radial and static) load is applied on pad (meaning on the pivot location) and between pad (usually symmetrically between two pivots), respectively. The use of these abbreviations “LOP” and “LBP” by the authors in this manuscript are often or always not correct. For example, the sentence “At lower eccentricities, the different between the LOP and LBP bearing coefficients are insignificant.” is not correct. In fact, in the context, “at lower eccentricities” means “at low orbit amplitudes”, “the different” should be “the difference”, “the LOP and LBP bearing coefficients” should be “the bearing dynamic coefficients are almost independent on the load direction” (or something similar). In addition, as there is no static loading, the load direction is always changing from 0 to 360 deg. like the angular location of the masse “m” (unbalance mass).
Response 31: Noted and updated accordingly.
Line 458 “For lower unbalance magnitude and rotor speed, however, the shape of the orbits looks relatively circular. This is because the bearings operate at low orbit amplitudes, and the bearing coefficients are almost independent of the journal radial position.”
Point 32: At the beginning, it should be clearly specified that there is no static radial load applied to the bearings, only a dynamic loading (synchronous unbalance load or pure rotating load) is considered in this study.
Response 32: Noted and updated accordingly.
Line 93 “No static radial load was considered, and the bearings were subjected for a dynamic load due to a rotating unbalance.”
Point 33: A nomenclature with a clear definition and units of all variables and parameters used in this manuscript is needed.
Response 33: Noted and updated accordingly.
Point 34: The authors wrote this first sentence in the last section “5. Discussions and Conclusions”: “The steady-state and non-stationary responses of the vertical rotor with TPJBs were successfully modelled by predefining the bearing coefficients using polynomial equations.” No result and no discussion on steady-state regime have been presented in this manuscript, only some dynamic coefficients have been calculated for given (imposed) eccentricities.
Response 34: Noted and updated accordingly.
The “steady-state response simulation” has been changed to “response simulation under constant rotor spin speed”
The “non-stationary response simulation” has been changed to “response simulation under variable rotor spin speed” or “transient”
Point 35: The authors wrote this sentence in the last section “5. Discussions and Conclusions”: “These equations were derived from the two-dimensional polynomial regression, and the bearing coefficients were represented as functions of eccentricity ratio and rotor speed.” Note that the dynamic coefficients of the bearing are also dependent of the angle “alpha”, or of the angular location of the journal center, not only of journal eccentricity.
Response 35: Noted and updated accordingly.
Line 533 “The dynamic responses of the vertical rotor with TPJBs under constant and variable rotor spin sped were successfully modelled by predefining the bearing coefficients using polynomial equations and periodic functions (sine and cosine). Theses equations represents the direct and cross-coupling bearing coefficients as functions of the location of the journal centre and rotor speed.”.
Point 36: The authors wrote this sentence in the last section “5. Discussions and Conclusions”: “Steady-state response simulations were carried out for different unbalance magnitudes and rotor speeds, …” The steady-state regime is obtained for a constant radial load, constant in both magnitude and direction leading to a unique journal center position. When an unbalance loading is applied, the journal center location is varying: it describes a closed orbit or a trajectory and so, it could be a cycling regime (dynamic regime) or a transient regime (when the rotational speed is varying, for example) but it is not at all a steady-state regime (permanent regime).
Response 36: Noted and updated accordingly. See (34)
Point 37: The authors wrote this sentence in the last section “5. Discussions and Conclusions”: “Since the bearing properties were different at LOP and LBP, the four pad bearing generated square-shaped orbits and forces at large unbalance magnitudes.” However, for the above reason (see comment 31), “different at LOP and LBP” is not correct and not appropriate in this sentence; please, correct it. It is also better to write “the four-shoes tilting pad journal bearing” instead of “the four pad bearing”. This result “generated square-shaped orbits” is not a new result as it was previously obtained by numerous other researchers: a triangle-shape orbit by Desbordes for the three-shoes tilting-pad journal bearing and a square-shape orbit by Monmousseau for the four-shoes tilting-pad journal bearing, for examples.
Response 37: Noted and updated accordingly. The following sentence has been removed from the manuscript
“Since the bearing properties were different at LOP and LBP, the four pad bearing generated square-shaped orbits and forces at large unbalance magnitudes.”
Point 38: The authors wrote this sentence in the last section “5. Discussions and Conclusions”: “… the difference between the bearing coefficients at LOP and LBP were insignificant.”. This is not correct, see my previous comment (31).
Response 38: Noted and updated accordingly.
Line 545 “At lower unbalance magnitudes and rotor speeds, however, the bearings operated at low orbit amplitude, and the bearing dynamic coefficients are almost independent on the load direction.”
Point 39: The authors wrote this sentence in the last section “5. Discussions and Conclusions”: “The square-shaped displacement orbits …”. This is not correct, see my previous comment (23).
Response 39: Noted and updated accordingly.
Point 40: General comment: all the text, the captions and the tables should be checked in order to correct the manuscript by considering all the above comments. I cannot spend all my time by checking and correcting all mistakes/errors/ incorrectness.
In conclusion, a serious, major and strong revision of the manuscript is requested following the above mandatory comments/recommendations.

Reviewer 2 Report
Bearing coefficients used in the paper are calculated for a constant eccentricity and a constant rotor speed. These coefficients were determined using a commercial software. Coefficients determined by commercial software are calculated under the assumption that shaft centre is not moving in bearing (no orbit). Therefore, the influence of moving shaft (orbit) is not considered in the calculation of bearing coefficients. This simplification makes fast computation possible, but is not accurate. Other published models do consider this effect. Please include a discussion about it. Probably the good agreement between experimental data and simulation is because this effect is not important in the laboratory test rig presented in this paper, but for an industrial machine seems to be fundamental.
In the introduction, different previous approaches were mentioned. We suggest to include, for example, the use of predefined bearing forces as a function of eccentricity, change in time of eccentricity, rotating speed, angle of inclination and change in time of inclination presented by Yu Huang et al and used in a vertical actual machine in Perez et al.
Yu Huang, Zhuxin Tian, Runchang Chen, Haiyin Cao, 2017. A simpler method to calculate instability threshold speed of hydrodynamic journal bearing. Mechanism and Machine Theory, 108, 209-216.
Perez, N., Rodriguez, C. 2020. Vertical Rotor Model with Hydrodynamic Journal Bearings. Engineering Failure Analysis, 119
Author Response
Response to Reviewer 2 Comments
Dear Reviewer,
First of all, we would like to thank you for your very interesting comments and feedbacks. The authors tried to answer the comments as clear and as comprehensive as possible. Based on the reviewer’s feedback, some more explanations are included in the manuscript. Please let us know if there is anything unclear or needs more clarifications.
Further changes on the manuscript:
New viscosity value of the lubricant (i.e. 46 mPa.s) has been used in FEM of Model II. The value was taken from a commercial software (RAPPID) for Model I, which uses Vogel’s law and calculate the oil viscosity for a given temperature. As a result, Figure 15, Figure 16, and Table 5 may look slightly different from those in the first draft of the manuscript. No significant change has been observed on the computational time (Figure 15 & Table 5) and bearing force (Figure 16 & Table 5), although the trajectory of the journal center increased by around 12%. However, the change does not affect the conclusion or discussion of the manuscript as the main objective is to compare the computational efficiency of the two models.
The followings are notes and explanations for the review comments and feedbacks.
Point 1: Bearing coefficients used in the paper are calculated for a constant eccentricity and a constant rotor speed. These coefficients were determined using a commercial software. Coefficients determined by commercial software are calculated under the assumption that shaft centre is not moving in bearing (no orbit). Therefore, the influence of moving shaft (orbit) is not considered in the calculation of bearing coefficients. This simplification makes fast computation possible, but is not accurate. Other published models do consider this effect. Please include a discussion about it. Probably the good agreement between experimental data and simulation is because this effect is not important in the laboratory test rig presented in this paper, but for an industrial machine seems to be fundamental.
Response 1: Noted and the following sentences are added to the manuscript.
Line 161: “The bearing coefficients were calculated at given eccentricities, assuming no whirling of the journal. A quasi-static approximation was applied, and the whirling effect was not considered when solving the fluid-film lubrication model. In other words, a journal’s tangential velocity () was assumed to be dependent only on the spinning speed of the journal. For small orbit amplitudes (e), the journal’s tangential velocity due to whirling effect can be ignored. For large orbit amplitudes (compared to the radius of the journal), however, the velocity component due to the moving journal center cannot be ignored, and quasi-static approximation could be inadequate. In this paper, the amplitude of the whirl was considerably smaller than the radius of the rotor, and the dynamic effect of the moving journal center was neglected. Thus, the quasi-static approximation is reasonable. Similarly, for large hydropower rotors, the radius ratio (e/R) is even significantly small, and the quasi-static approximation can be adequate.”
Figure. The schematic of rotor spin and whirl.
Point 2: In the introduction, different previous approaches were mentioned. We suggest to include, for example, the use of predefined bearing forces as a function of eccentricity, change in time of eccentricity, rotating speed, angle of inclination and change in time of inclination presented by Yu Huang et al and used in a vertical actual machine in Perez et al.
Yu Huang, Zhuxin Tian, Runchang Chen, Haiyin Cao, 2017. A simpler method to calculate instability threshold speed of hydrodynamic journal bearing. Mechanism and Machine Theory, 108, 209-216.
Perez, N., Rodriguez, C. 2020. Vertical Rotor Model with Hydrodynamic Journal Bearings. Engineering Failure Analysis, 119
Response 2: Note and the above references are included in the manuscript.
Line 63-67 “Many attempts have been made by researchers to simplify and improve the computation efficiency of the simulation procedure. Perez et al. [28] used a simplified analytical bearing force expression to model the dynamics of a vertical rotor with hydrodynamic journal bearing. This approach has previously been presented by Yu Huang et al. [29] to estimate the instability threshold speed of cylindrical journal bearings.”
Round 2
Reviewer 1 Report
The authors have considered the reviewer's comments. I have no further comments.
Reviewer 2 Report
Thank you for the modifications